# Contrasting effects of Western vs Mediterranean diets on monocyte inflammatory gene expression and social behavior in a primate model

Corbin SC Johnson[1], Carol A Shively[2], Kristofer T Michalson[2], Amanda J Lea[3,4], Ryne J DeBo[2], Timothy D Howard[5], Gregory A Hawkins[5], Susan E Appt[2], Yongmei Liu[6], Charles E McCall[7], David M Herrington[8], Edward H Ip[9], Thomas C Register[2]†*, Noah Snyder-Mackler[1,10,11,12,13]†*

[1]Department of Psychology, University of Washington, Seattle, United States; [2]Department of Pathology, Section on Comparative Medicine, Wake Forest School of Medicine, Winston-Salem, United States; [3]Lewis-Sigler Institute for Integrative Genomics, Princeton University, Princeton, United States; [4]Department of Ecology and Evolutionary Biology, Princeton University, Princeton, United States; [5]Department of Biochemistry, Wake Forest School of Medicine, Winston-Salem, United States; [6]Division of Cardiology, Duke University School of Medicine, Durham, United States; [7]Department of Internal Medicine, Section of Molecular Medicine, Wake Forest School of Medicine, Winston-Salem, United States; [8]Department of Internal Medicine, Section on Cardiovascular Medicine, Wake Forest School of Medicine, Winston-Salem, United States; [9]Department of Biostatistics and Data Science, Wake Forest School of Medicine, Winston-Salem, United States; [10]Center for Studies in Demography and Ecology, University of Washington, Seattle, United States; [11]Department of Biology, University of Washington, Seattle, United States; [12]School of Life Sciences, Arizona State University, Tempe, United States; [13]Center for Evolution & Medicine, Arizona State University, Tempe, United States

*For correspondence:
register@wakehealth.edu (TCR);
nsnyderm@asu.edu (NS-M)

†These authors contributed equally to this work

Competing interests: The authors declare that no competing interests exist.

**Abstract** Dietary changes associated with industrialization increase the prevalence of chronic diseases, such as obesity, type II diabetes, and cardiovascular disease. This relationship is often attributed to an 'evolutionary mismatch' between human physiology and modern nutritional environments. Western diets enriched with foods that were scarce throughout human evolutionary history (e.g. simple sugars and saturated fats) promote inflammation and disease relative to diets more akin to ancestral human hunter-gatherer diets, such as a Mediterranean diet. Peripheral blood monocytes, precursors to macrophages and important mediators of innate immunity and inflammation, are sensitive to the environment and may represent a critical intermediate in the pathway linking diet to disease. We evaluated the effects of 15 months of whole diet manipulations mimicking Western or Mediterranean diet patterns on monocyte polarization in a well-established model of human health, the cynomolgus macaque (*Macaca fascicularis*). Monocyte transcriptional profiles differed markedly between diets, with 40% of transcripts showing differential expression (FDR < 0.05). Monocytes from Western diet consumers were polarized toward a more proinflammatory phenotype. The Western diet shifted the co-expression of 445 gene pairs, including small RNAs and transcription factors associated with metabolism and adiposity in humans, and dramatically altered behavior. For example, Western-fed individuals were more

anxious and less socially integrated. These behavioral changes were also associated with some of the effects of diet on gene expression, suggesting an interaction between diet, central nervous system activity, and monocyte gene expression. This study provides new molecular insights into an evolutionary mismatch and uncovers new pathways through which Western diets alter monocyte polarization toward a proinflammatory phenotype.

## Introduction

Modern human diets vary across geography, cultures, and socioeconomic strata and have profound impacts on health, survival, and reproduction. The Western diet, prevalent in high-income countries, has been long associated with adverse effects on health, particularly in relation to chronic diseases of aging (*Cordain et al., 2005*; *Drake et al., 2018*; *Jacka et al., 2010*; *Manzel et al., 2014*; *Pontzer et al., 2018*; *Smil, 1989*; *Smyth and Heron, 2006*). Western diets are high in simple sugars and saturated and n-6 fatty acids, which increase sympathetic nervous activity, oxidative stress, and levels of inflammatory markers (*Drescher et al., 2019*; *Giugliano et al., 2006*; *Holt et al., 2009*; *Lopez-Garcia et al., 2004*; *Nanri et al., 2007*; *Nettleton et al., 2006*). Consequently, Western diets are associated with increased risk for metabolic syndrome (*Drake et al., 2018*), type II diabetes (*Smyth and Heron, 2006*), cardiovascular disease (*Drake et al., 2018*; *Smil, 1989*), nonalcoholic hepatosteatosis (*Jump et al., 2015*), autoimmune disorders (*Manzel et al., 2014*), depression (*Jacka et al., 2010*), and premature death (*Cordain et al., 2005*). From an evolutionary perspective, the negative health effects of Western diets are hypothesized to be driven by a 'mismatch' between human physiology – which evolved to subsist on a plant-based diet supplemented with fish and meat but no refined products – and the radically different nutritional environment of many human populations today (*Eaton et al., 1988*; *Lieberman, 2014*; *Stearns and Koella, 2008*).

In contrast to the Western diet, the Mediterranean diet derives most protein and fat from vegetable sources, which are enriched with antioxidants, monounsaturated and n-3 fatty acids. This diet more closely resembles that of modern hunter-gatherer populations and presumed ancestral human populations in macronutrient composition and key dietary components (*Mackenbach, 2007*; *Pontzer et al., 2018*). Interestingly, the Mediterranean diet is also associated with an anti-inflammatory phenotype (*O'Keefe et al., 2008*), reduced incidence of chronic disease, and increased longevity, relative to a Western diet (*Farchi et al., 1994*; *Osler and Schroll, 1997*; *Romagnolo and Selmin, 2017*; *Trichopoulou et al., 1995*). At face value, the detrimental health effects associated with Western relative to Mediterranean diets are consistent with evolutionary mismatch. However, the mechanisms through which this mismatch may negatively and causally affect health, and conversely how the Mediterranean diet positively impacts health remains poorly understood. Disentangling these mechanisms is especially difficult in humans, as population shifts toward Western diets may be accompanied by other challenges to health such as reduced physical activity or increased total caloric intake (*Snodgrass, 2013*; *Kraft et al., 2018*; *Lagranja et al., 2015*).

One potential mechanism for dietary impacts on health is through changes to our immune system. Previous attempts to understand how Western versus Mediterranean diets impact the immune system have relied on correlational analyses of self-reported diet or short-term dietary interventions in humans, which are limited in their ability to address causality (*Stice and Durant, 2014*; *Suchanek et al., 2011*). Many experimental manipulations have focused on single nutrients in animal models (*Hu, 2002*; *Kimmig and Karalis, 2013*; *Ohlow et al., 2017*; *Steinhubl, 2008*; *Whelton et al., 1992*), which cannot address the potentially important synergistic effects of the multiple nutrients that make up human diet patterns. Our study design employs whole diet manipulations in a randomized preclinical trial framework (Western versus Mediterranean) to address the role that monocytes play in sensing and responding to dietary inputs (*Devêvre et al., 2015*; *Drescher et al., 2019*; *Holt et al., 2009*; *Nanri et al., 2007*; *Nettleton et al., 2006*). Monocytes and monocyte-derived macrophages are innate immune cells that vary phenotypically along a spectrum, which ranges broadly from proinflammatory (M1-like) to regulatory/reparative (M2-like). An appropriate balance of these monocyte phenotypes is essential for a healthy immune system. Classically-activated M1 monocytes respond to proinflammatory cytokines such as tumor necrosis factor (TNF)-$\alpha$ and interferon (IFN)-$\gamma$ by becoming macrophages, which propagate the inflammatory response toward infection (*Mosser and Edwards, 2008*). In contrast, M2 activated monocytes

mobilize the tissue repair processes and release anti-inflammatory cytokines in response to IL-4, IL-13, and transforming growth factor (TGF)-β (*Mosser and Edwards, 2008*). Thus, dietary constituents or patterns may influence pathologic processes by altering the balance between these proinflammatory and anti-inflammatory monocyte subsets – a hypothesis that has yet to be tested (*Devêvre et al., 2015*).

In addition to diet, psychosocial stress is also known to impact immune phenotypes. In particular, multiple sources of social adversity, such as low social status and poor social integration, have been shown to increase the expression of inflammatory genes in primary white blood cells in humans and other animals (*Cole, 2013*; *Cole, 2019*; *Cole et al., 2015*; *Snyder-Mackler et al., 2016*; *Snyder-Mackler and Lea, 2018*; *Tung and Gilad, 2013*). Given that some food constituents can directly alter social behaviors themselves (*Hollis et al., 2018*; *Kaplan et al., 1991*; *Kasprowska-Liśkiewicz et al., 2017*; *Kougias et al., 2018*; *Warden and Fisler, 2008*), it is therefore possible that diet effects on immune cell regulation may, to some degree, be mediated through changes in these behaviors. It is also possible that diet-induced alterations in systemic inflammation may alter behavior. However, because no detailed studies of diet, social behavior, and immune cell phenotypes have been conducted, it remains unclear how these factors are linked and how, together, they impact health.

To overcome the limitations of human studies, we designed a randomized preclinical trial in cynomolgus macaques (*Macaca fascicularis*), a well-established model of dietary and behavioral influences on health in which we can carefully control diet and the environment. Macaques are excellent models for human health and disease as they share many core genetic, physiological, and behavioral phenotypes with humans (*Jarczok et al., 2018*; *Kromrey et al., 2016*; *Shively, 1998*; *Shively and Day, 2015*; *Willard and Shively, 2012*). In this study, we conducted a whole-diet manipulation to directly and simultaneously compare the behavioral and physiological effects of Mediterranean and Western diets, formulated to mimic human diet patterns. The randomized trial design allowed us to identify causal effects of realistic, complex diet patterns on one possible mechanism linking diet to chronic disease risk – polarization of immune cell populations toward a proinflammatory phenotype. Previous reports from this preclinical trial demonstrate that relative to the Mediterranean diet, the Western diet increased body weight, body fat, insulin resistance, and hepatosteatosis (*Shively et al., 2019*); exacerbated autonomic and hypothalamic-pituitary-adrenal responses to psychosocial stress (*Shively et al., 2020*); and altered brain neuroanatomy (*Frye et al., 2021*). Here, we report the effects of the Mediterranean and Western diet patterns on behavior and monocyte gene expression.

## Results

### Diet intervention

Adult female cynomolgus macaques were fed either a Western-like (hereafter, 'Western', n = 20) or a Mediterranean-like (hereafter, 'Mediterranean', n = 15) diet for 15 months (approximately equivalent to four human years). The experimental diets were formulated to model human diet patterns and have been previously described (*Shively et al., 2019*). Briefly, the Western diet was designed to mimic the diet typically consumed by middle-aged Americans (*Centers for Disease Control and Prevention (CDC), 2014*), whereas the Mediterranean diet reflected key aspects of the human Mediterranean diet (*Kafatos et al., 2000*). The experimental diets were matched on macronutrients and cholesterol content but differed in fatty acids. Fats and proteins were mostly plant based in the Mediterranean diet (*Kafatos et al., 2000*), and from animal sources in the Western diet. This resulted in high levels of monounsaturated fats in the Mediterranean diet, and saturated fats in the Western diet (*Cordain et al., 2005*; *Kafatos et al., 2000*). The Mediterranean diet was higher in complex carbohydrates and fiber, and had a lower n-6:n-3 fatty acid ratio (similar to a modern-day, traditional hunter-gatherer type of diet [*Cordain et al., 2005*]), and lower sodium and refined sugars than the Western diet. Key Mediterranean ingredients included English walnut powder and extra-virgin olive oil which were the primary components provided to participants in the PREDIMED trial (*Estruch et al., 2018*). Macronutrient composition of experimental diets compared to monkey chow and human diet patterns can be found in *Table 1*, Methods.

**Table 1.** Comparison of nutritional contents of diet patterns in human with nonhuman primate diets used in the current study.

| Diet Composition | Human | | Nonhuman Primate | | |
| --- | --- | --- | --- | --- | --- |
| | Western | Mediterranean | Western[*] | Mediterranean[*] | Chow[†] |
| **% of Calories** | | | | | |
| Protein | 15[§] | 17[¶] | 16[§] | 16[¶] | 18 |
| Carbohydrate[‡] | 51[§] | 51[¶] | 54[§] | 54[¶] | 69 |
| Fat | 33[§] | 32[¶] | 31[§] | 31[¶] | 13 |
| **% of Total fats** | | | | | |
| Saturated | 33[§] | 21[¶] | 36[§] | 21[¶] | 26 |
| Monounsaturated | 36[§] | 56[¶] | 36[§] | 57[¶] | 28 |
| Polyunsaturated | 24[§] | 15[¶] | 26[§] | 20[¶] | 32 |
| **Other nutrients** | | | | | |
| ω6:ω3 Fatty Acids | 15:1[††] | 2.1-3:1[‡‡] | 14.8:1[††] | 2.9:1[‡‡] | 12:01 |
| Cholesterol mg/Cal | 0.13[§] | 0.16[¶] | 0.16[§] | 0.15[¶] | trace |
| Fiber g/Cal | 0.01[§] | 0.03[§§] | 0.02[§] | 0.04[§§] | 0.01 |
| Sodium mg/Cal | 1.7[§,¶¶] | 1.3[¶,§§] | 1.7[§,¶¶] | 1.1[¶,§§] | 0.25 |

[*] Developed and prepared at Wake Forest School of Medicine.

[†] LabDiet Chemical Composition Diet 5037/8. Type of fat known in 86% of total fat. Omega-6 from corn and pork fat.

[‡] Human carbohydrate calories include alcohol.

[§] (**Centers for Disease Control and Prevention (CDC), 2014**).

[¶] (**Bédard et al., 2012**).

[††] (**Simopoulos, 2006**).

[‡‡] (**Cordain et al., 2005**).

[§§] (**Kafatos et al., 2000**).

[¶¶] (**Powles et al., 2013**).

Reprinted from **Shively et al., 2019**, *Obesity* with permission (**Shively et al., 2019**).

## Diet induced major shifts in monocyte gene expression

RNA sequencing was employed to measure genome-wide gene expression of purified CD14+ monocytes after 15 months on the experimental diets. Diet had a strong effect on monocyte gene expression: the first principal component of the correlation matrix of normalized residual gene expression (see Materials and methods), which explained 59% of the overall variance, was significantly associated with diet ($t_{(25.1)}$ = 4.4, $p$ = 1.7 x $10^{-4}$; *Figure 1A*). PC1 score was correlated with expression of known proinflammatory genes such as interleukin-6 (*IL6* Pearson's $r$ = 0.77, $p$ = 5.4 x $10^{-8}$), interleukin-1α (*IL1A* Pearson's $r$ = 0.69, $p$ = 4.3 x $10^{-6}$), and two subunits of the NF-κB protein (*NFKB1* Pearson's $r$ = 0.61, $p$ = 1.2 x $10^{-4}$; *NFKB2* Pearson's $r$ = 0.72, $p$ = 1.3 x $10^{-6}$). Approximately 40% of the 12,240 tested genes were significantly differentially expressed genes (DEGs) between the two diets ($n$ = 4900 genes, FDR < 0.05; for all detected genes and the effect size of diet, see *Supplementary file 1A*; for DEGs sorted by the effect size of diet, see *Supplementary file 1B*). The number of diet-responsive genes was roughly balanced between those that were more highly expressed in monkeys fed the Mediterranean diet ($n$ = 2,664; hereafter 'Mediterranean genes') and those that were more highly expressed in monkeys fed the Western diet ($n$ = 2,236; hereafter 'Western genes'). While balanced in direction, the effect sizes of diet in Western genes were, on average, 1.6-fold larger than in Mediterranean genes (Mann-Whitney $U$ = 4.1 x $10^{6}$, $p$ = 6.1 x $10^{-117}$; *Figure 1B*). Thus, the strongest effects were observed in genes that were either activated by a Western diet or suppressed by a Mediterranean diet.

## Functional characterization of differentially expressed genes

Monocytes from animals fed the Western diet had higher expression of a number of well-known inflammatory-related genes, including *IL6* ($\beta_{diet}$ = 1.63, FDR = 0.025; *Figure 1B*), *IL1A* ($\beta_{diet}$ = 1.22, FDR = 0.033), and two subunits of the NF-κB protein (*NFKB1* $\beta_{diet}$ = 0.30, FDR = 0.017; *NFKB2* $\beta_{diet}$

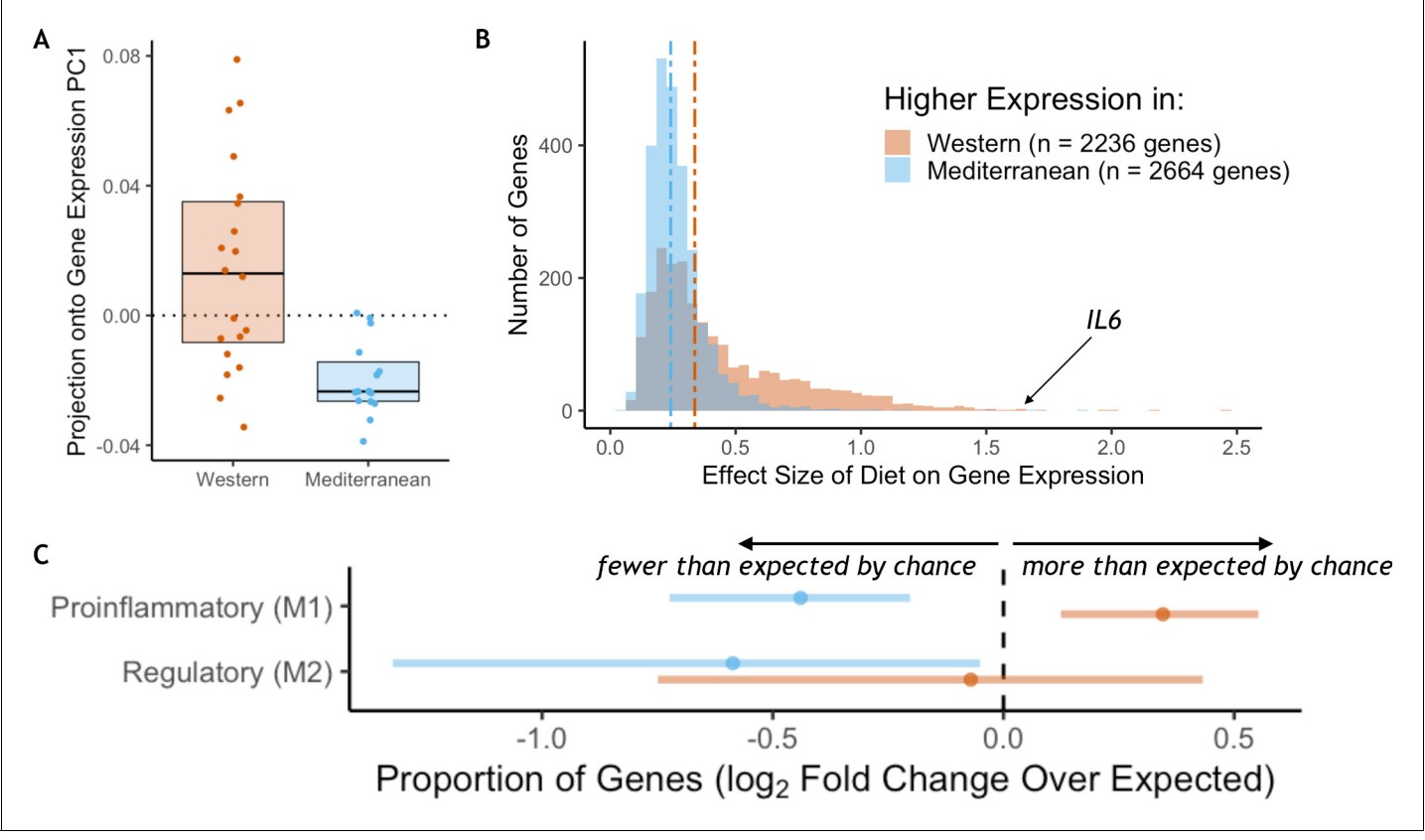

**Figure 1.** Diet effects on monocyte gene expression. (**A**) Diet was significantly associated with the first principal component of gene expression (59% variance explained, $t_{(25.1)}$ = 4.4, $p$ = 1.72 x 10$^{-4}$). (**B**) The average effect size of diet on Western genes was 60% stronger than the effect size of diet on Mediterranean genes (Mann-Whitney $U$ = 4.1 x 10$^6$, $p$ = 6.1 x 10$^{-117}$). (**C**) Western genes (orange) contained more M1 genes than expected by chance, indicating that the Western diet induced a shift toward a proinflammatory monocyte phenotype. Western genes were enriched for proinflammatory (M1-like) genes (fold-enrichment = 1.27, *95% CI* = 1.09, 1.46), while Mediterranean genes (blue) were depleted of these same M1-like genes (fold-enrichment = 0.74, *95% CI* = 0.61, 0.88). Regulatory (M2-like) genes were also under-represented in Mediterranean genes (fold-enrichment = 0.67, *95% CI* = 0.40, 0.97), but not in Western genes (fold-enrichment = 0.95, *95% CI* = 0.60, 1.35). *Figure 1—figure supplement 1*. The sets of Western and Mediterranean genes were compared to genes implicated in 103 complex human diseases and traits (*Zhang et al., 2020*). Fisher's Exact Tests were used to calculate the enrichment of trait-associated genes in Western genes that are depicted here (FDR < 0.02, 95% CI) and no traits were enriched in Mediterranean genes. *Figure 1—figure supplement 2*. Western-diet fed animals exhibited significantly higher expression of pro-inflammatory genes involved in the conserved transcriptional response to adversity (CTRA *Cole et al., 2015*; Mann-Whitney $U$ = 222, $p$ = 0.016) and lower expression of antiviral- and antibody-related CTRA genes (Mann-Whitney $U$ = 82, $p$ = 0.023), both consistent with the CTRA. See *Supplementary file 1A* for CTRA categories.

The online version of this article includes the following figure supplement(s) for figure 1:

**Figure supplement 1.** Genes with significantly higher expression in monkeys fed the Western diet ('Western genes') were enriched for genes associated with numerous complex human diseases and traits.

**Figure supplement 2.** Expression of genes in the conserved transcriptional response to adversity (CTRA *Cole et al., 2015*) indicate inflammatory effects of a Western diet that parallel the effects of social adversity.

= 0.42, FDR = 0.012). Western genes were more likely to be involved in replication and metabolic cellular processes, including response to growth factor (GO:0070848, weighted Fisher's Exact Test (FET) $p$ = 4.6 x 10$^{-3}$) and response to insulin (GO:0032868, weighted FET $p$ = 4.0 x 10$^{-4}$), suggesting that the Western diet also reprogrammed oxidative metabolic aspects of monocyte gene regulation. Conversely, Mediterranean diet monocyte expression patterns were involved in enhanced oxidation-reduction processes (GO:0055114, weighted FET $p$ = 6.0 x 10$^{-3}$), a critical function in keeping proinflammatory monocytes in check (for all GO terms enriched in Western and Mediterranean genes, see *Supplementary file 1A-B*). When compared to genes causally implicated at the expression level in 103 complex human diseases and traits (*Zhang et al., 2020*), we found that

Western genes were enriched for genes involved in multiple human diet-associated diseases and traits (celiac disease: fold enrichment = 1.80, $p$ = 0.016; body fat: fold enrichment = 0.26, $p$ = 2.9 x $10^{-3}$; and body mass index: fold enrichment = 0.20, $p$ = 0.016; *Figure 1—figure supplement 1*), as well as genes associated with levels of important metabolites such as HDL cholesterol (fold enrichment = 0.61, $p$ = 6.8 x $10^{-3}$), LDL cholesterol (fold enrichment = 0.63, $p$ = 0.012), and adiponectin (fold enrichment = 1.32, $p$ = 7.7 x $10^{-3}$). In contrast, Mediterranean genes were not enriched for any of the 103 complex traits tested (all FDR > 0.2).

We next conducted a more targeted analysis of monocyte polarization by focusing on genes previously shown to be differentially expressed between induced proinflammatory (M1) and regulatory (M2) monocytes (*FANTOM consortium et al., 2014*) (see *Supplementary file 1A* for polarization categories). Western genes contained more M1-associated genes than expected by chance ($n$ = 162 genes, fold-enrichment = 1.27, *95% CI* = 1.09, 1.46; *Figure 1C*), but not M2-associated genes ($n$ = 24 genes, fold-enrichment = 0.95, *95% CI* = 0.60, 1.35). Conversely, both M1-associated genes ($n$ = 112 genes, fold-enrichment = 0.74, *95% CI* = 0.61, 0.88) and M2-associated genes ($n$ = 20 genes, fold-enrichment = 0.67, *95% CI* = 0.40, 0.97) were underrepresented among Mediterranean genes.

## Association of transcription factors with differentially expressed genes

To identify putative upstream gene regulatory mechanisms, we examined whether DEGs were associated with predicted *cis*-regulatory transcription factor binding sites. We identified 34 distinct transcription factor-binding motifs enriched within 2 kilobases of the transcription start sites of Mediterranean genes and one that was enriched near the transcription start sites of Western genes (FDR < 0.05; *Figure 2*, for all transcription factor binding motifs enriched in the regulatory regions of either set of diet genes, see *Supplementary file 1*). Diet altered expression of the genes encoding for seven of these 35 transcription factors, including *IRF3*, *IRF8*, *MEF2C*, and *SP1*, which drive monocyte fate and polarization in response to extracellular signals (*Chistiakov et al., 2018*; *Günthner and Anders, 2013*; *Schüler et al., 2008*; *Scott et al., 1994*; *Zhang et al., 1994*). Thus, some of the diet-associated changes in monocyte transcriptional profiles may be mediated by changes in the expression and *cis*-regulatory binding of these key transcription factors.

## Gene co-expression modules recapitulate functional role of diet-induced changes

We employed a commonly used bioinformatic approach, weighted gene co-expression network analysis (WGCNA) (*Langfelder and Horvath, 2008*) to group genes by pattern of transcription into co-expression modules. Overall, we identified 15 modules of co-expressed genes. Module 5 was more highly expressed in Mediterranean-fed animals (Welch-Satterthwaite $t_{(28.3)}$=−3.9, Holm-Bonferroni-adjusted $p$ ($p_{HB}$) = 8.1 x $10^{-3}$; see *Supplementary file 1A* for all co-expression modules), and was depleted for M1 genes ($n$ = 383 genes, log odds ratio = −0.6, *95% CI* = −0.8,–0.3, $p_{HB}$ = 5.2 x $10^{-5}$). Similar to the set of Mediterranean genes, module 5 was enriched for genes involved in the oxidation-reduction process (GO:0055114, weighted FET $p$ = 3.1 x $10^{-7}$; for all GO terms passing an adjusted p-value threshold of 0.05, see *Supplementary file 1B*). While only one module was associated with higher expression in the Mediterranean diet, we found two modules, modules 8 and 10, that were more highly expressed in Western-fed animals (module 8: $t_{(32.3)}$=3.5, $p_{HB}$ = 0.020; module 10: $t_{(33.0)}$=3.1, $p_{HB}$ = 0.048). These two modules exhibited similar gene regulatory signatures, as both modules were enriched for regulation of transcription by RNA polymerase II (module 8: GO:0006357 (overall regulation), weighted FET $p$ = 1.3 x $10^{-5}$; module 10: GO:0045944 (positive regulation), weighted FET $p$ = 3.1 x $10^{-7}$). Module 10 also included more M1 genes than expected ($n$ = 186, log odds ratio = 0.8, *95% CI* = 0.5, 1.0, $p_{HB}$ = 1.9 x $10^{-7}$). A third module that trended toward higher expression in the Western diet, module 9 ($t_{(32.2)}$=2.5, raw $p$ = 0.019, $p_{HB}$ = 0.19), was enriched for genes involved in the inflammatory response (GO:0006954, weighted FET $p$ = 2.4 x $10^{-6}$). Together, these results reinforce our findings that the Western diet contributes to proinflammatory polarization in a multi-faceted manner, while the Mediterranean diet can contribute to the reduction of oxidative stress. Interestingly, two modules, modules 4 and 12, were depleted for diet-associated genes (module 4: FET $p$ = 2.1 x $10^{-17}$, module 12: FET $p$ = 3.9 x $10^{-19}$) and were enriched for genes involved in the defense response to virus (module 4: GO:0051607, weighted FET $p$ = 1.1 x $10^{-18}$) and the adaptive immune response (module 12: GO:0002250, weighted FET $p$ =

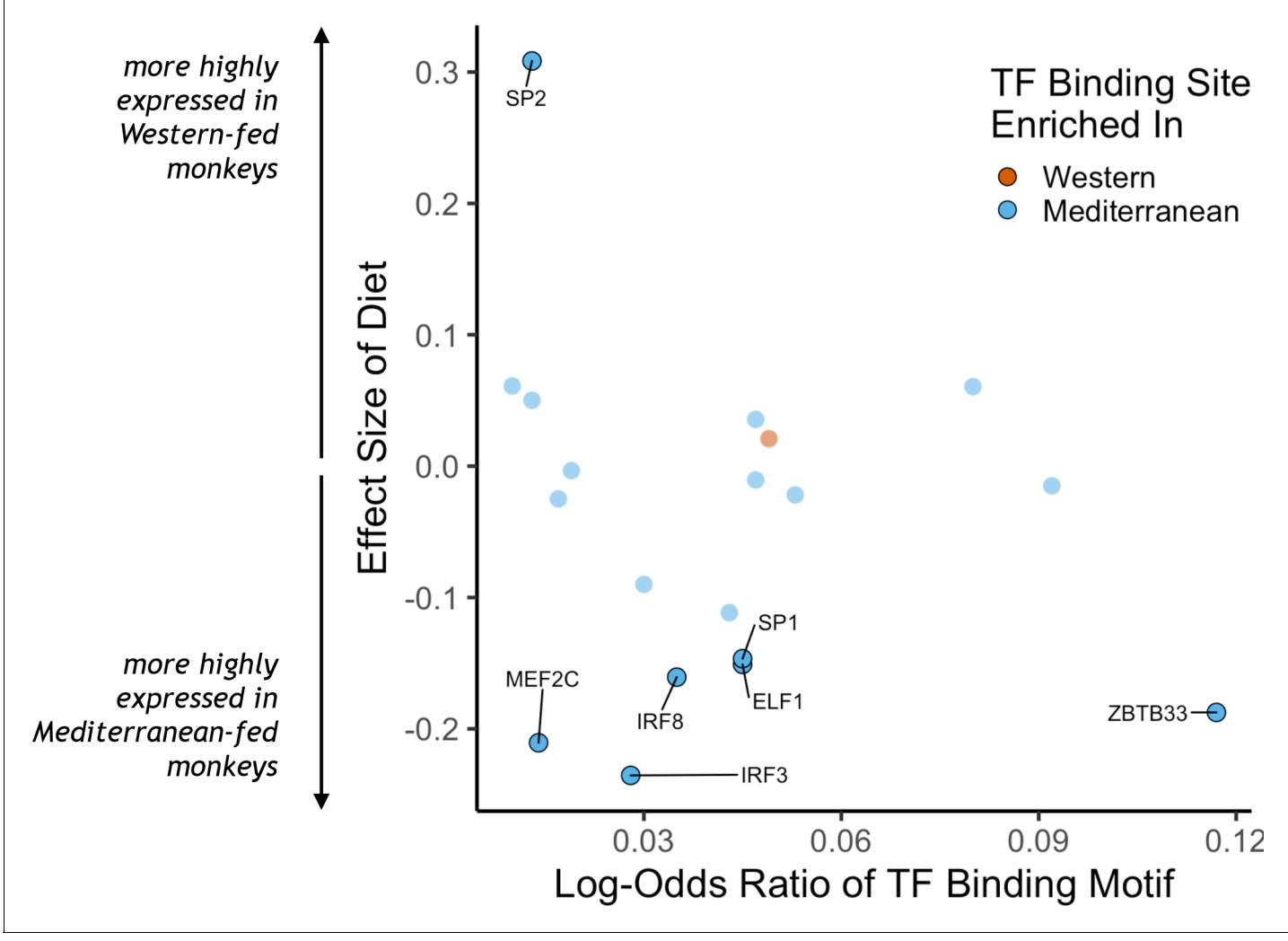

**Figure 2.** Transcription factor (TF) binding motifs correlated with diet effects on gene expression. The log-odds ratio of TF binding motif enrichment in Western genes (orange) or Mediterranean genes (blue) are depicted on the x-axis. The y-axis shows the effect size of diet on the expression of the gene that encodes for the TF. Only TFs with binding motifs significantly enriched in either gene set and that were detectably expressed in our samples are shown, with those significantly affected by diet (FDR < 0.05) outlined and labeled.

$8.1 \times 10^{-9}$). This suggests that viral responses and adaptive immunity may be less affected by the diet.

## Diet altered gene co-expression patterns

The effects of diet on the magnitude and direction of pairwise gene expression correlations were assessed for the most strongly diet-affected genes, as such effects could reveal key gene regulatory networks that are altered by diet, that may themselves be regulated by key upstream targets (*de la Fuente, 2010*; *Gaiteri et al., 2014*). To reduce the number of tests, we limited our analyses to the pairwise combinations of the top 140 DEGs ($n$ = 9730 combinations). Of these gene pairs, many were significantly associated with each other in both diets, both positively ($n$ = 714) and negatively ($n$ = 332, $p < 0.05$; for all gene pairs tested and their correlations, see *Supplementary file 1—5A*), suggesting that while diet altered expression of these genes, it did not change their co-expression relationships. Drawing on a newly developed approach, 'correlation by individual level product' (CILP) (*Lea et al., 2019*), we identified 445 other gene pairs that exhibited significant differences (FDR < 0.2) in their correlation between the Mediterranean- and Western-fed monkeys

(*Supplementary file 1—5A*; *Figure 3A*), suggesting that one of the experimental diets altered the coherence between the genes (*Figure 3A*).

We also identified 16 'hub' genes that exhibited differential correlations with more partner genes than expected by chance (*Figure 3B*, for all genes included in one or more differentially correlated gene pairs, see *Supplementary file 1—5B*). These hub genes were enriched for genes encoding transcription factors (OR = 7.40, FET $p$ = 7.0 x 10$^{-3}$), including *SOX4* (essential for normal insulin secretion and glucose tolerance) and *NR4A2* (involved in lipid, carbohydrate, and energy metabolism *Goldsworthy et al., 2008*; *Pearen and Muscat, 2010*), providing further support for immunological and metabolic reprogramming induced by our diet manipulation. Interestingly, the hub gene involved in the greatest number of differentially-correlated gene pairs was *RF00283, aka SCARNA18*, a non-coding RNA that has been associated with BMI, HDL cholesterol, and aging in human genome-wide association studies (*Davis et al., 2017*; *Dluzen et al., 2018*; *Kanai et al., 2018*; *Tachmazidou et al., 2017*; *Figure 3B–D*). This small nucleolar RNA is thus a key transcriptional regulator that is altered by diet and has a cascading effect on other genes and pathways.

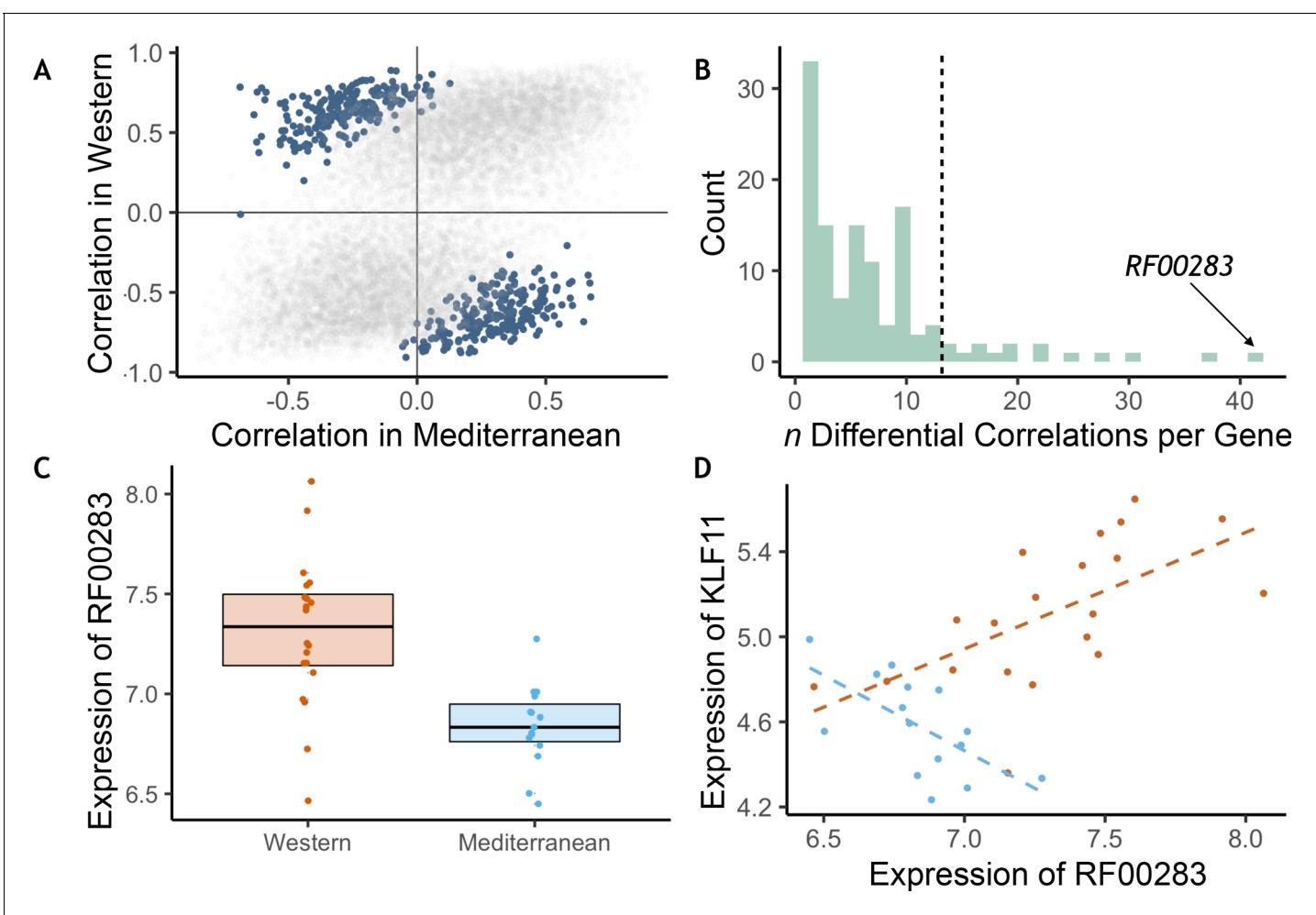

**Figure 3.** Diet altered monocyte gene co-expression. (A) The Pearson correlation between each pair of genes within each of the experimental diets. Gene pairs that were significantly differently correlated between diets are highlighted in blue ($n$ = 445 significant pairs, FDR < 0.2). (B) Of the genes involved in significant pairs, some were paired with more genes than expected by chance, called 'hub' genes ($n$ = 16 hub genes; dotted black line is the maximum number of significant pairs expected by chance). The strongest hub gene was the non-coding RNA *RF00283*. (C) Residual normalized expression of *RF00283* is significantly greater in Western- than Mediterranean-fed monkeys ($\beta_{diet}$ = 0.51, FDR = 2.3 x 10$^{-6}$). (D) Example of a differential correlation involving *RF00283*. Residual normalized expression of *RF00283* is plotted against expression of *KLF11*, a differentially-expressed transcription factor that regulates insulin and has been associated with type II diabetes in humans (*Neve et al., 2005*). The two genes were more highly expressed in Western monocytes, were positively correlated with one another in Western-fed monkeys (Pearson's $r$ = 0.61, $p$ = 4.2 x 10$^{-3}$), were negatively correlated in Mediterranean-fed monkeys (Pearson's $r$ = −0.63, $p$ = 0.011), and were differentially correlated between the two diets ($p$ = 4.1 x 10$^{-5}$, FDR = 0.11).

## Diet altered social and affective behavior

In order to understand how diet may impact behavior and how both may interact to impact health, behavioral data were collected weekly during two 10 min focal observations. These data were collected during both the baseline (2 hr/monkey total) and experimental phases (mean = 17.6 hr/monkey total) of the study. There were no significant differences in behavior between assigned diet groups during the baseline phase while consuming chow (*Figure 4—figure supplement 1A–B*). However, after 15 months on experimental diets, the two diet groups differed significantly in behavior. The Mediterranean group spent more time in body contact (Mann-Whitney $U = 284$, Holm-Bonferroni-adjusted $p$ ($p_{HB}$) = 1.1 x $10^{-5}$) and resting ($U = 269$, $p_{HB}$ = 1.6 x $10^{-3}$), while those fed the Western diet spent more time alone ($U = 255$, $p_{HB}$ = 4.9 x $10^{-3}$ *Figure 4A*; see *Figure 4—figure supplement 1C–D* for behaviors during experimental diet consumption).

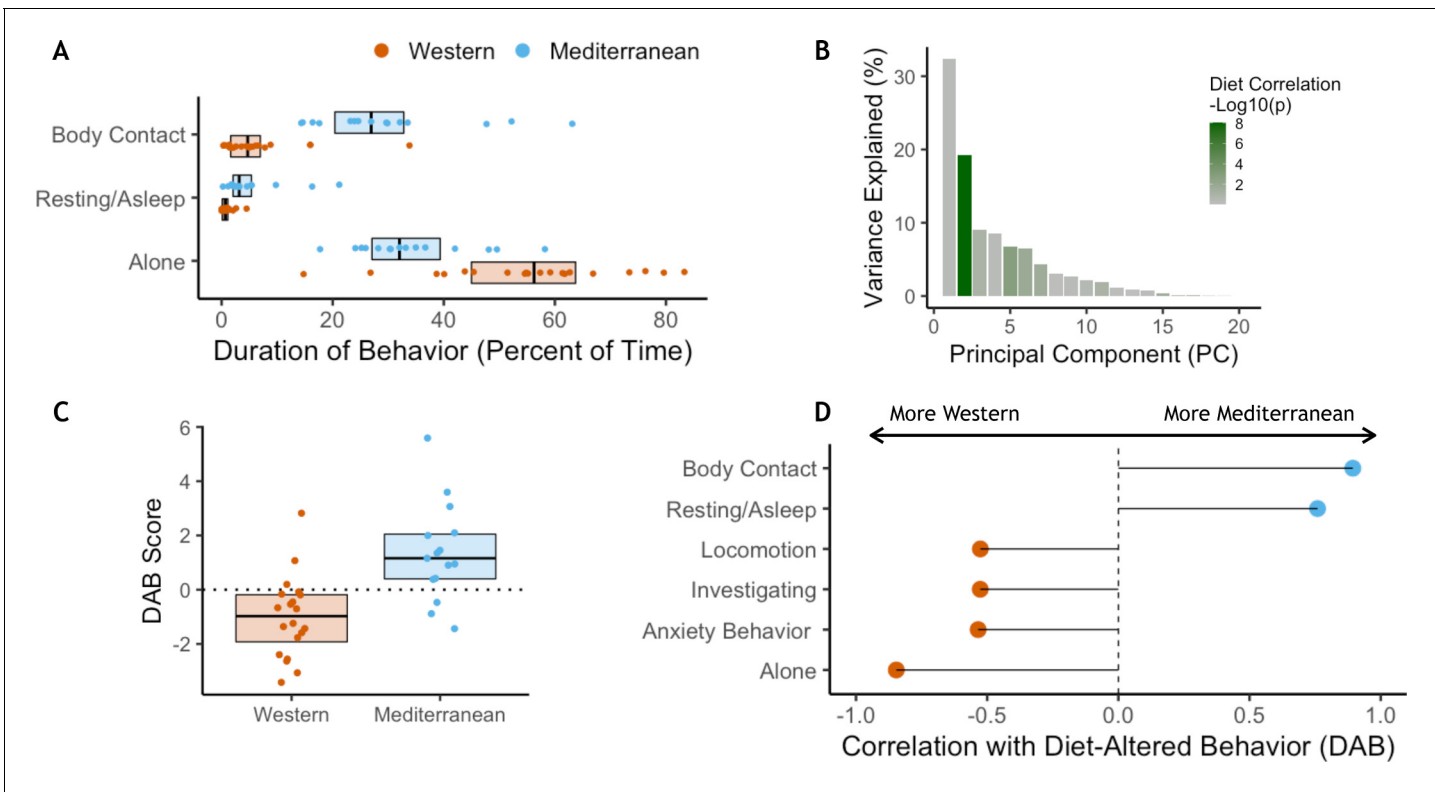

**Figure 4.** Diet alters behavioral phenotype. (A) Three behaviors were significantly different between the two diet groups. Monkeys fed the Mediterranean diet spent more time in body contact (Holm-Bonferroni adjusted $p$ ($p_{HB}$) = 1.1 x $10^{-5}$) and resting ($p_{HB}$ = 1.6 x $10^{-3}$) than Western-fed monkeys. Monkeys eating the Western diet spent more time alone than Mediterranean-fed monkeys ($p_{HB}$ = 4.9 x $10^{-3}$). (B) Principal component 2 (PC2) explained 19% of the variance in behavior and was the only PC significantly correlated with diet. (C) PC2 represents a composite measure of diet-altered behavior, as individual loadings onto PC2 ('DAB scores'; 19% of all variance in behavior) were significantly higher in Mediterranean diet compared to Western diet animals ($t_{(26.8)}$ = 4.13, $p$ = 3.2 x $10^{-4}$). (D) Six of the 20 behaviors observed are significantly correlated with DAB score ($p_{HB}$ < 0.05). Here, significant correlations with DAB score in which behaviors are more frequent in Mediterranean diet or Western diet monkeys are indicated with blue or orange points, respectively. *Figure 4—figure supplement 1*. There were no differences between the Western- and Mediterranean-fed groups in the rates (A) or duration (B) of behaviors during the baseline phase, prior to diet manipulation. The boxplots depict the per-group medians and interquartile ranges for each behavior. Animals fed the Western diet are colored orange, and those fed the Mediterranean diet colored blue. Significant differences between the diet groups in the rates (C) or duration (D) of behaviors during the experimental phase are indicated (Mann-Whitney U test, Holm-Bonferroni adjusted p < 0.05 *, p < 0.01 **, p < 0.001 ***). *Figure 4—figure supplement 2*. The first axis of variance in behavior—which explained 31% of the overall variance—was significantly positively correlated with dominance rank across diets (Pearson's $r$ = 0.84, $p$ = 3.9 x $10^{-10}$). All monkeys are assigned a rank between 0 and 1 based on the outcomes of dyadic interactions, where a higher rank indicates more dominant social status.

The online version of this article includes the following figure supplement(s) for figure 4:

**Figure supplement 1.** Diet manipulation altered behavior.

**Figure supplement 2.** The first PC of all behavioral data captures dominance rank.

Principal component analysis was conducted to identify key behaviors associated with one another (**Benito et al., 2018**; **Seltmann et al., 2018**). Behaviors associated with dominance interactions—including aggression, submission, and being groomed—all loaded heavily onto the first principal component, which explained 32% of the overall variance in behavior and did not differ between diets (Welch-Satterthwaite $t_{(30.4)} = -0.3$, $p = 0.70$; for relationship between dominance rank and PC1, see **Figure 4—figure supplement 2**; for further discussion of social status in these animals, see Appendix 1).

The second principal component explained 19% of the variance in behavior (**Figure 4B**) and differed significantly between the two diets ($t_{(26.8)}=4.1$, $p = 3.2 \times 10^{-4}$; **Figure 4C**). No other principal component of behavioral phenotypes was significantly correlated with diet (**Figure 4B**). PC2 captured socially relevant behaviors that also differed between the diets and thus represents a composite of diet-altered behaviors (hereafter DAB). Specifically, DAB score (i.e. an individual's PC2 projection) was positively correlated with percent of time spent in body contact, indicative of social integration (Pearson's $r = 0.89$, $p_{HB} = 1.0 \times 10^{-11}$; **Figure 4D**), and higher in Mediterranean-fed animals. Conversely, percent of time spent alone was associated with lower DAB scores (Pearson's $r = -0.85$, $p_{HB} = 3.0 \times 10^{-9}$), and was higher in animals fed the Western diet. Previous work has validated a behavioral index of anxiety in nonhuman primates (rate of self-grooming and scratching) (**Coleman et al., 2011**; **Maestripieri et al., 1992**; **Schino et al., 1996**; **Shively et al., 2015**; **Troisi et al., 2000**; **Troisi, 2002**), which loaded heavily onto PC2 and is significantly negatively correlated with DAB score (Pearson's $r = -0.53$, $p_{HB} = 0.019$). Thus, PC2 (DAB) captured a measure of social integration associated with consuming a Mediterranean-like diet, and social isolation and anxiety associated with consuming a Western-like diet.

## Diet-altered behaviors and monocyte gene expression as mediators

Given the effects of diet on both behavior and gene expression, we used mediation analyses to explore the potential influences of one on the other. Of the 4900 DEGs, 29% were also significantly associated with the DAB score in a univariate model ($n = 1,414$, FDR < 0.05). Of these, DAB score significantly mediated the effect of diet on the expression of 1199 genes (24% of all DEGs, $p < 0.05$; **Figure 5A**). Among these DAB-mediated genes, DAB score mediation accounted for significantly more of the total effect of diet in Western genes (mean = 52.6%, s.d. = 12.6%), than Mediterranean genes (mean = 45.3%, s.d. = 10.1%; Mann-Whitney $U = 1.1 \times 10^5$, $p = 6.4 \times 10^{-25}$; **Figure 5B**). These DAB-mediated genes were also significantly more likely to be Western genes than Mediterranean genes ($n = 712$ Western genes, 59%, two-sided binomial test $p = 1.5 \times 10^{-21}$), and were enriched in regulation of inflammatory response (GO:0050727, weighted FET $p = 2.9 \times 10^{-3}$; for all GO terms significantly enriched in DAB-mediated genes, see **Supplementary file 1—6A-C**). Together, these observations suggest that the effect of diet on monocyte gene regulation may partially be due to diet-induced changes in key social behaviors.

We also tested the hypothesis that peripheral immune cell gene expression mediated the effects of diet on behavior in the 27% of DEGs for which monocyte gene expression significantly predicted DAB in a univariate model ($n = 1324$, FDR < 0.05). Gene expression mediated the effect of diet on DAB score in 898 genes (18% of all DEGs, $p < 0.05$; **Figure 5A**). Almost all of these genes (99%; 889/898) were in the set of genes for which behavioral changes mediated changes in gene expression. The genes that mediated the effect of diet on DAB score were more likely to be Western genes ($n = 523$ Western genes, 58%, two-sided binomial test $p = 4.6 \times 10^{-14}$); however, the portion of the total effect of diet that was accounted for by gene expression did not vary between Western (mean = 27.1%, s.d. = 5.2%) and Mediterranean genes (mean = 27.1%, s.d. = 4.5%; Mann-Whitney $U = 1.0 \times 10^5$, $p = 0.55$; **Figure 5C**).

## Diet differentially induced expression of the conserved transcriptional response to adversity (CTRA) genes

Additional analyses focused on expression of a well-studied set of social adversity-responsive genes known as the 'conserved transcriptional response to adversity' (CTRA) (**Cole et al., 2015**) in the Western- and Mediterranean-fed animals in our study. Animals fed a Western diet exhibited significantly higher expression of pro-inflammatory genes included in the CTRA (Mann-Whitney $U = 222$, $p = 0.016$) and lower expression of antiviral- and antibody-related CTRA genes (Mann-Whitney $U = 82$,

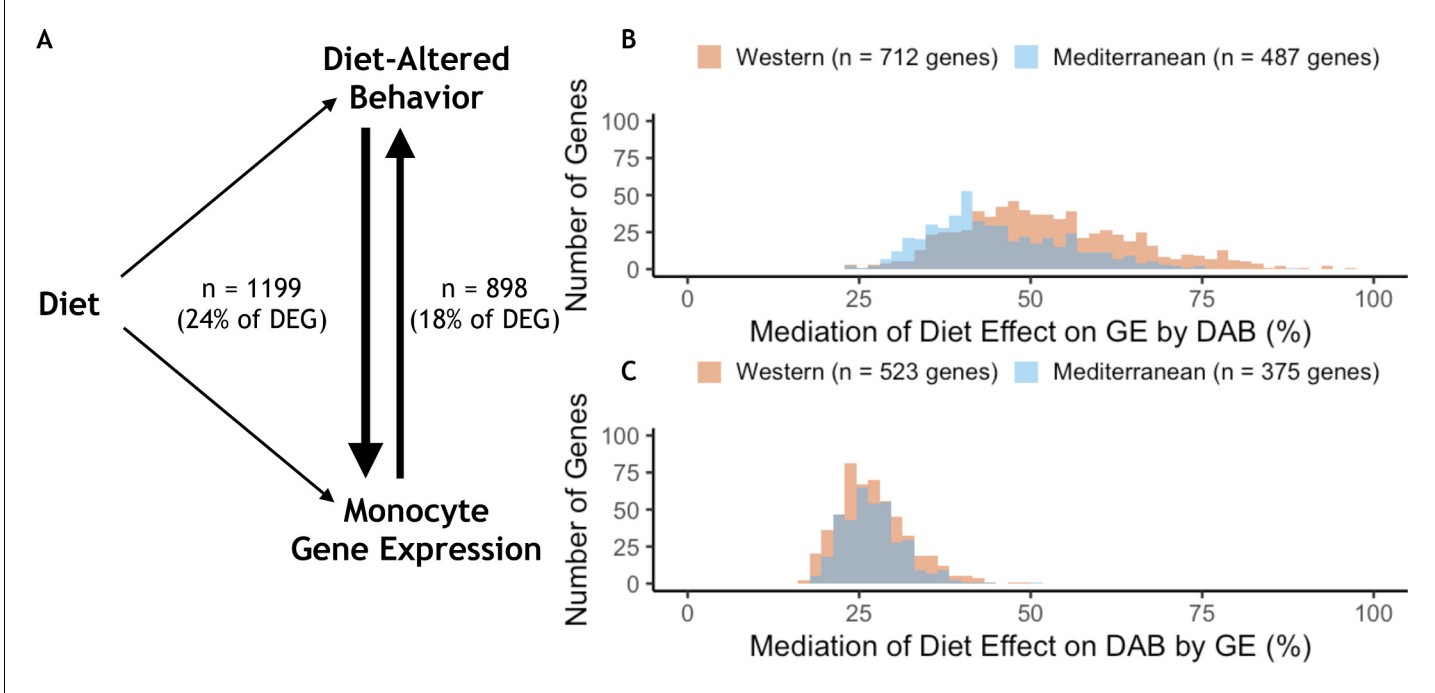

**Figure 5.** Behavior partially mediates the effect of diet on gene expression for 24% of diet-associated genes. (**A**) Diet-altered behavior (DAB) mediated the effect of diet on gene expression for 24% (n = 1199) of genes for which diet had an effect (differentially expressed genes or DEGs). For 18% of DEGs, gene expression mediated the effect of diet on DAB score. (**B**) DAB score mediated 23–97% of the total effect of diet on gene expression in 1199 genes (n = 712 Western genes, orange; n = 487 Mediterranean genes, blue). DAB score mediated a greater number of Western genes than Mediterranean genes (p = 1.5 x 10$^{-21}$) and accounted for a greater portion of the effect size of diet (p = 6.4 x 10$^{-25}$) in Western genes. (**C**) In gene-by-gene models of DAB score as a function of diet + gene expression, gene expression mediated 17–51% of the total effect of diet on DAB in 898 genes (n = 523 Western genes; n = 375 Mediterranean genes). Gene expression mediated a greater number of Western genes than Mediterranean genes (p = 4.6 x 10$^{-14}$), although expression of these genes did not account for more of the effect of diet on DAB score than Mediterranean genes (Mann-Whitney U = 1.0 x 10$^{5}$, p = 0.55).

p = 0.023; *Figure 1—figure supplement 2*; for categorization of CTRA genes, see *Supplementary file 1—1A*).

### Western diet induced a mosaic response

Western diet induced substantial variation in multiple phenotypes, including body weight, gene expression, and behavior; consistent with previous studies demonstrating that some individuals may be more resistant (or susceptible) to the effects of a Western diet (*Shively et al., 2009*), presumably due to genetic variation or past environmental exposures. However, we were unable to identify consistencies in individual responsiveness across the phenotypes (*Appendix 2—figure 1*). For instance, monkeys that exhibited a strong gene regulatory response to the Western diet did not necessarily exhibit a large increase in body weight or a strong negative DAB score (all p > 0.2). Furthermore, change in body weight did not significantly predict gene expression in monocytes (all FDR > 0.2). Western diet fed individuals thus exhibited a mosaic response to diet across multiple phenotypes, presumably involving interactions between diet, stress, behavior, environment, microbiome, and genome/epigenome.

### Discussion

This study shows, for the first time, that a whole-diet manipulation exerted profound effects on monocyte polarization and social behavior in primates. Forty percent of monocyte-expressed genes were differentially expressed between monkeys fed Western or Mediterranean diets, indicating that diet dramatically altered monocyte programming. Relative to monocytes from Mediterranean-fed subjects, monocytes from Western diet consumers exhibited increased expression of

proinflammatory and monocyte polarization regulatory genes. Our findings extend previous studies, such as a randomized human cross-over trial that demonstrated changes in monocyte proinflammatory genes, including *IL6*, other interleukins, and NF-κB components, in elderly individuals consuming a Mediterranean like diet enriched in olive oil versus a diet more enriched in saturated fat (*Camargo et al., 2012*).

We identified a putative molecular mechanism, altered monocyte polarization, that may contribute to the established links between changes in human diets associated with industrialization and increases in chronic disease (*Cordain et al., 2005*; *Drake et al., 2018*; *Jacka et al., 2010*; *Lea et al., 2020*; *Manzel et al., 2014*; *Pontzer et al., 2018*; *Smil, 1989*; *Smyth and Heron, 2006*). Comparative studies of human health across different modern populations – namely those consuming traditional hunter-gatherer, forager-horticulturalist, or pastoralist diets versus modern, Western-like diets – lend support for the evolutionary mismatch hypothesis (*Eaton et al., 1988*; *Kaplan et al., 2017*; *Lea et al., 2020*; *Pontzer et al., 2018*). In particular, this work has found that traditional populations have much lower rates of non-communicable diseases, especially cardiometabolic diseases, relative to Western societies; however, because so many lifestyle factors differ between traditional societies and those in Western, high income countries, it has been difficult to understand the role of diet specifically in driving health variation or to address causality (*Snodgrass, 2013*; *Kraft et al., 2018*; *Lagranja et al., 2015*). Additionally, it is difficult to collect samples appropriate for genomic analyses from subsistence-level groups, and consequently the molecular correlates of industrial transitions and evolutionary mismatch remain largely unexplored. Our preclinical randomized study design allows us to draw causal inferences about the role of Western diets in the development of chronic diseases of aging, and provides important data about cellular and molecular mechanisms that may contribute to evolutionary mismatch. These data set the stage for future studies that could compare the transcriptional response to diet in our preclinical study with gene regulatory variation observed between traditional and more market-integrated or Western-like human groups.

Beyond changes in gene expression, we also identified differences in gene co-expression and enrichment of transcription factor binding motifs, suggesting that diet exerts differential effects on gene regulatory networks. Many transcription factors appear to be involved in diet-regulated gene expression. Members of the E26 transformation-specific (ETS), specificity protein (Sp)/Krüppel-like family (KLF), myocyte-specific enhancer factor (MEF), and interferon-regulatory factor (IRF) families of transcription factors, which have all been linked to myeloid differentiation (*Chistiakov et al., 2018*; *Schüler et al., 2008*; *Scott et al., 1994*; *Zhang et al., 1994*), were overrepresented in regulatory regions of genes with higher expression in the Mediterranean diet group ('Mediterranean genes'). Three IRF family transcription factors had binding motifs enriched in Mediterranean genes: IRF-1 and IRF-8 are both linked to M1 monocyte polarization, while IRF-3 is associated with M2 polarization. The sole transcription factor with binding sites enriched in Western diet-associated genes, ATF2, is a key mediator of inflammatory pathways and diseases, including response to bacterial endotoxin, atherosclerosis, and obesity (*Fledderus et al., 2007*; *Miyata et al., 2012*; *Reimold et al., 2001*). Western genes were enriched for activation of the MAPKK pathway, which lies upstream of ATF2 (*Herlaar and Brown, 1999*), supporting a role in monocyte polarization. Transcription factors were also overrepresented in the pairs of differentially co-expressed genes, further indicating that diet alters regulatory networks and monocyte differentiation and polarization.

It is also worth pointing out that changes in gene co-expression and network connectivity have been previously proposed as a response to novel or challenging environmental conditions, including Western diets. In particular, work on decanalization has hypothesized that gene regulatory networks evolve over many generations of stabilizing selection, and that novel environmental challenges (such as Western diets and lifestyles) may disrupt these fine-tuned connections leading to dysregulation, a breakdown in co-expression, and ultimately disease (*Careau et al., 2014*; *Gibson, 2009a*; *Gibson, 2009b*; *Hu et al., 2016*; *Lea et al., 2019*). In support of this idea, we found diet-induced changes in the co-expression of transcription factors involved in insulin secretion and glucose tolerance (SOX4), lipid, carbohydrate, and energy metabolism (NR4A2), and BMI, HDL, and aging (*RF00283*) (*Davis et al., 2017*; *Dluzen et al., 2018*; *Goldsworthy et al., 2008*; *Kanai et al., 2018*; *Pearen and Muscat, 2010*). We also observed that the transcription factor *MEF2D*, which has previously been implicated in the transcriptomic response to insulin signaling (*Samson and Wong, 2002*; *Solomon et al., 2008*), is a hub gene identified in 22 differentially-correlated gene pairs. Hub genes

like *MEF2D* may pinpoint optimized systems that break down as a result of mismatch and are thus intriguing targets for future analyses.

It is worth noting that the dichotomous M1/M2 paradigm of monocyte polarization is an oversimplification of the more complex heterogeneity of monocytes. (*Martinez and Gordon, 2014*; *Nahrendorf and Swirski, 2016*) For example, there are at least three classes of monocytes in the circulation–classical, intermediate, and non-classical. We did not assess the relative abundance of these subsets, thus the observed gene expression patterns could reflect either changes in the relative proportions of these subsets and/or shifts in monocyte polarization within subsets (*Michalson et al., 2019*; *Wolf et al., 2017*).

The diets also altered key behaviors. Monkeys consuming the Western diet exhibited more behaviors related to anxiety and social isolation, a phenotype remarkably similar to that observed in juvenile Japanese macaques born to mothers consuming a high-fat Western diet (*Thompson et al., 2018*). In that study, offspring behavior was associated with maternal levels of macrophage-derived chemokine (MDC), which showed higher expression in Western-diet fed animals in our study ($\beta_{diet}$ = 0.243, FDR = 0.059). Our findings suggest that a Western diet may also exert similar behavioral effects when consumed during adulthood.

There are myriad pathways through which diet may affect behavior. Diet may induce changes in the central nervous system (CNS) by altering gut microbiota which alters vagal input to the brain (*Bonaz et al., 2018*). Previous results from our study demonstrated a strong diet effect on the gut microbiome (*Nagpal et al., 2018a*), and lower parasympathetic (vagal) activity in the Western diet group at the time the monocyte transcriptome was assessed (*Shively et al., 2020*). Taken together these observations suggest that diet-induced changes in vagal tone in the gut-brain axis may be one pathway through which diet impacted brain function, potentially affecting behavior.

Diet-altered behaviors were linked to changes in monocyte gene expression. For a subset (24%) of genes, the DAB score mediated the effect of diet on monocyte gene expression. Monocytes have been shown to be responsive to social isolation (*Cole, 2019*) and anxiety (*Cole et al., 2015*). Social isolation and anxiety, produced by Western diet consumption, may be accompanied by increased sympathetic outflow and increased hypothalamic-pituitary adrenal production of cortisol, both of which modulate monocyte intracellular processes governing inflammatory molecule production (*Cacioppo et al., 2015*; *Holwerda et al., 2018*; *Juruena et al., 2020*). Supporting the involvement of these systems, we previously reported that the Western diet group had increased sympathetic activity, and increased cortisol concentrations (*Shively et al., 2020*). Western diet may contribute to inflammation by producing a more socially isolated or anxious animal with increased sympathetic and hypothalamic pituitary adrenal activity, which in turn alters monocyte function. Higher expression of genes in the conserved transcriptional response to adversity support this pathway (*Cole, 2019*; *Cole et al., 2015*; *Figure 1—figure supplement 2*). Behavior is a functional assay for the CNS. Thus, this observation suggests that diet may alter CNS function, which may in turn alter circulating monocyte gene expression.

In a somewhat smaller and overlapping subset of genes (18%), diet-induced differences in monocyte gene expression significantly mediated the effect of diet on behavior (DAB). This observation suggests that diet alters monocyte gene expression, which in turn may affect CNS function. There are a variety of mechanisms through which diets differentially influence the nervous system. Western diet may disrupt the blood-brain barrier, increasing infiltration of Western-diet induced cytokines, chemokines, and myeloid cells from the periphery (*Raison et al., 2006*; *Yang et al., 2019*). Once in the brain these molecules can alter BDNF production, neurotransmitter systems, and hypothalamic-pituitary-adrenal function (*Raison et al., 2006*). Western diet induced inflammatory molecules also may affect the brain through direct effects on the afferent vagus nerve (*Maier and Watkins, 1998*), activation of glial cells (*Graham et al., 2016*), or alteration of neuronal membrane lipid composition affecting neurotransmission (*Du et al., 2016*), whereas a Mediterranean diet may have direct anti-inflammatory actions by increasing n-3 fatty acids in the brain (*Layé et al., 2018*).

Together, these results support both mediation pathways, suggesting that multiple mechanistic pathways may contribute to these observations; however, we are unable to conclusively state that one mediation pathway is supported over the other or delineate the exact role of the CNS. As each gene is modeled independently in the mediation analyses, it is possible that the expression of a subset of genes in monocytes alters CNS function and induces behavioral change, while expression of

another subset of genes is responsive to behavioral phenotypes and/or CNS function. These potential pathways present intriguing possibilities for future experiments.

Monkeys fed the Western diet displayed a heterogeneous response to the diet manipulation across physiological (e.g. body weight), gene regulatory, and behavioral measures. Rather than a single pattern of response to diet where the physiological changes are predictive of behavioral or gene regulatory changes in response to diet, there was no correlation between these measures in monkeys fed the Western diet. This suggests that physiological changes such as weight gain may not be the primary link between diet, poor immune function, and negative health consequences. Understanding both behavioral and gene regulatory responses to environmental mismatch, such as those introduced by dietary patterns, will help to understand the subsequent impact on health.

It is important to note the strengths and limitations of the current study. Macaques continue to be a critical model for understanding human health and disease, including on the influence of diet on numerous phenotypes including atherosclerosis and cardiovascular disease, bone metabolism, breast and uterine biology, and other physiological and pathological phenotypes (*Adams et al., 1997*; *Clarkson et al., 2004*; *Clarkson et al., 2013*; *Cline et al., 2001*; *Cline and Wood, 2006*; *Haberthur et al., 2010*; *Lees et al., 1998*; *Mikkola et al., 2004*; *Mikkola and Clarkson, 2006*; *Naftolin et al., 2004*; *Nagpal et al., 2018a*; *Nagpal et al., 2018b*; *Register, 2009*; *Register et al., 2003*; *Shively and Clarkson, 2009*; *Sophonsritsuk et al., 2013*; *Walker et al., 2008*; *Wood et al., 2007*). In a publication based on the same study animals, dietary manipulation produced changes in the gut microbiome similar to that seen in humans consuming Western and Mediterranean diets (*Nagpal et al., 2018b*), which further supports the translational relevance of our findings with diet-induced changes in gene expression in genes involved in human health and disease. Nevertheless, extrapolation of the current findings to human health should be done with caution, as the last common ancestor of humans and macaques lived 25 million years ago and evolution has shaped the physiology and natural diet of each species in distinct ways (*Luca et al., 2010*). A related complication is the difficulty in defining a 'control' diet for both human and nonhuman primates. Macaques in the wild are omnivorous, and standard monkey chow derives most of its protein content from soy, which is rich in isoflavones such as genistein and daidzein known to have biological activity (*Zaheer and Humayoun Akhtar, 2017*). Thus, in the context of the evolutionary mismatch hypothesis, standard monkey chow does not recapitulate a natural macaque diet that could serve as a control for the current diet manipulation, leaving the interpretation of the current results ambiguous as to which diet is driving the changes observed in one diet group relative to the other, which could be addressed in future studies.

In summary, we found that diet significantly alters monocyte polarization and gene expression, and to a lesser extent behavior. The Western diet promoted a proinflammatory monocyte phenotype relative to a Mediterranean diet, which supports the role of monocyte polarization in diet-associated chronic inflammatory diseases. Thus, altered monocyte programming could represent one mechanism underlying an evolutionary mismatch between our past and current diets. The majority of the effects of diet are presumably mediated through direct or combined actions of saturated/polyunsaturated fats, n-6:n-3 ratios, pro- and anti-antioxidant characteristics, and other features of the Western diet inconsistent with the nutritional environment in which humans and nonhuman primates evolved. Ongoing and future work will address interactions between social behavior (e.g. social status) and diet to further understand how environmental stressors may impact inflammation in the periphery and in the central nervous system.

## Materials and methods

### Subjects

Forty-three adult (age: mean = 9.0, range = 8.2–10.4 years, estimated by dentition), female cynomolgus macaques (*Macaca fascicularis*), were obtained (Shin Nippon Biomedical Laboratories, USA SRC, Alice, TX) and housed at the Wake Forest School of Medicine Primate Center (Winston-Salem, NC) (*Shively et al., 2019*). Briefly, the monkeys were socially housed in groups of 3–4 and consumed standard monkey chow (*Table 1*) during an eight-month baseline phase, after which pens were assigned to receive either the Western (five groups, *n* = 21) or Mediterranean (six groups, *n* = 22) diet, balanced on pretreatment characteristics that reflected overall health, including body weight,

body mass index, circulating basal cortisol, total plasma concentrations, and plasma triglyceride concentrations (*Shively et al., 2019*). Two monkeys did not tolerate the experimental diet, and were switched to standard monkey chow, three animals died during the course of the study (discussed in *Frye et al., 2021*), and three samples were removed for insufficient CD14 purification (see 'Removal of Batch Effects' below), resulting in a final sample size of 35 animals (Western *n* = 20, Mediterranean *n* = 15). All animal manipulations were performed according to the guidelines of state and federal laws, the US Department of Health and Human Services, and the Animal Care and Use Committee of Wake Forest School of Medicine.

## Experimental diets

Experimental diets (*Table 1*) were formulated to be isocaloric with respect to protein, fat, and carbohydrates, and identical in cholesterol content (~ 320 mg/2000 kilocalories (Cals)/day) as previously described (*Shively et al., 2019*).

## Behavioral characterization

Behavioral data were collected weekly during two 10 min focal observations, balanced for time of day, for 6 weeks during the baseline phase (2 hr/monkey total) and for 14 months during the experimental phase (mean = 17.6 hr/monkey total). Behaviors recorded included the frequency of aggressive and submissive behaviors, time spent in positive social interactions such as sitting in body contact and grooming or alone, and anxious behavior defined as self-directed behaviors including self-grooming and scratching (*Maestripieri et al., 1992*; *Schino et al., 1996*; *Shively et al., 2015*; *Troisi et al., 2000*; *Troisi, 2002*). Behaviors were collected as previously described (*Shively, 1998*), and combined into summary behaviors (e.g. 'aggression' was a combination of all total, noncontact, contact aggressive events). No significant differences in behavior were observed between the diet groups while consuming standard monkey chow diet during the baseline period (*Figure 4—figure supplement 1A,B*). In order to quantify the overall impact of diet on behavior, we conducted a principal component analysis using the R package *FactoMineR* (*Lê et al., 2008*). We corrected for multiple hypothesis tests using the Holm-Bonferroni adjusted p-values.

## Blood sample collection

The monkeys were trained to run out of their social groups on voice command. Blood was drawn via venipuncture within 9 min of entering the building. Blood was collected into EDTA-containing tubes, mixed with an equal amount of PBS without calcium or magnesium, and overlaid on a 90% Ficoll-Paque Plus/10% PBS solution in LeucoSep tubes followed by centrifugation at 800 x g for 20 min. Isolated PBMCs were then immediately used for the collection of CD14+ monocytes by positive selection using a Miltenyi bead-based protocol following manufacturer's instructions (Miltenyi Biotec, Bergisch Gladbach, Germany). After assessing cell viability and numbers, CD14+ monocytes were stored in 85% FBS, 15% DMSO sterile freezing media at −80°C and transferred to liquid nitrogen for storage until RNA extraction. Blood samples were collected from all subjects in a given social group on the same day and collection order was alternated between diets and randomized by group.

## RNA extraction and sequencing

RNA was extracted from monocytes using the AllPrep DNA/RNA Mini Kit (Qiagen, Inc, Hilden, Germany), and quantified using a NanoDrop spectrophotometer and Agilent 2100 Bioanalyzer with RNA 6000 Nano chips (Agilent Technology, Inc, Santa Clara, CA). RNA libraries were prepared for sequencing by the Cancer Genomics Shared Resource (Wake Forest School of Medicine, Winston-Salem, NC) using the TruSeq-stranded total RNA kit (Illumina), which includes a ribosomal depletion step. The RNA-seq libraries were then sequenced using single-end 76 bp reads on an Illumina Next-Seq 500 to an average read depth of 34.5 million reads per sample (range 25.9–41.6 million reads). Reads were mapped to the *Macaca fascicularis* reference genome (Macaca_fascicularis_5.0, v 93, Ensembl) (*Kersey et al., 2018*; *Kinsella et al., 2011*) using HiSat2 (*Kim et al., 2015*) and then converted to a sample-by-gene read count matrix using featureCounts (*Liao et al., 2014*) (median = 38.0%; range 24.5–50.4% of reads mapped to exons). Sample processing order was randomized and where possible all samples were manipulated simultaneously so as to avoid introducing batch effects.

### Read count normalization and removal of batch effects

First, we removed genes with low expression (median reads per kilobase per million reads mapped < 1), which resulted in 12,240 genes for downstream analyses. We normalized read counts using the *voom* function of the R package *limma* (*Ritchie et al., 2015*). While investigating monocyte purity, three samples differed in CD3 gene expression from the rest by several orders of magnitude. We concluded that these samples were contaminated with CD3+ cells (i.e. inefficient CD14 purification, *Appendix 2—figure 2*) and excluded them from all analyses, leaving a final sample size of 35 monkeys ($n$ = 20 fed the Western diet, $n$ = 15 Mediterranean diet). To control for batch effects related to RNA quality and monocyte purity, we calculated the residual gene expression from a model of normalized gene expression as a function of CD14 expression, CD3 expression, RNA integrity, and RNA concentration. These residual gene expression values were used for all subsequent analyses.

### Modeling effect of diet on gene expression

In order to determine which genes were significantly affected by diet, we modeled the residual expression of each gene as a function of diet using a linear mixed effects model controlling for relatedness among monkeys using the R package *EMMREML* (*Akdemir and Godfrey, 2015*). Relatedness was estimated using the ngsRelate program (*Hanghøj et al., 2019*) with SNP genotypes inferred from the RNA-seq reads using bcftools mpileup (*Li et al., 2009*). We calculated an empirical false discovery rate (FDR) for each gene using a permutation-based approach (*Snyder-Mackler et al., 2016*), and report genes that passed at FDR < 0.05. To examine global patterns of variation in gene expression, we conducted principal component analysis on the correlation matrix of normalized residual gene expression using the *prcomp* function in R.

### Enrichment analyses

Gene ontology (GO) enrichment analyses were conducted using Fisher's Exact Tests and the *weight01* algorithm to test for enrichment implemented in the R package *topGO* (*Alexa and Rahnenfuhrer, 2019*). For a more targeted analysis of M1 and M2 specific genes, we identified a set of DEGs in our data set that were previously found to be involved in monocyte polarization (*FANTOM consortium et al., 2014*) (638 proinflammatory and 138 regulatory), which we used to explore monocyte polarization in the current study. We calculated the proportion of genes more highly expressed in the Mediterranean- and Western-fed animals in each polarization category and tested for significance using a permutation test ($n$ = 100,000 permutations). To compare the DEGs identified to genes implicated in human health, we utilized gene sets associated with 103 complex human traits and diseases identified by a prior study (*Zhang et al., 2020*). Fisher's Exact Tests were used to test for enrichment of these gene sets in our Western or Mediterranean DEGs.

### Transcription factor binding site analysis

We tested for enrichment of transcription factor binding motifs within 2 kb (upstream or downstream) of the transcription start sites of differentially expressed 'Western genes' or 'Mediterranean genes' (FDR < 0.05) using the program HOMER (*Heinz et al., 2010*) and equivalent regions around the transcription start sites of all genes expressed in these data as the background set for enrichment testing. We searched for known vertebrate transcription factor binding motifs and report the TF motifs passing a threshold of FDR < 0.05.

### Gene-gene co-expression analysis

In addition to testing whether diet led to mean differences in gene expression between Western and Mediterranean animals, we also tested whether diet impacted the correlation structure among expressed genes (i.e. gene co-expression). Specifically, we employed 'correlation by individual level product' (CILP) (*Lea et al., 2019*) analyses to test whether diet affected the magnitude or direction of pairwise gene expression correlations among the top 140 DEGs ($n$ = 9730 gene-gene pairs tested). To test whether a given pair of genes was differentially co-expressed as a function of diet, we first obtained a vector of products for each gene pair by multiplying the normalized gene expression values for two genes together. Normalization was performed by scaling expression values to mean 0 and unit variance within Mediterranean and Western subsets of the data respectively, to ensure that distributional differences between sample groups did not bias our results, following the

CILP authors' recommendations (*Lea et al., 2019*). Each of these vectors of products were used as the outcome variable in a linear mixed effects model implemented in the R package *EMMREML* (*Akdemir and Godfrey, 2015*), which included a fixed effect of diet and a random effect to control for genetic relatedness. To assess significance, we extracted the p-value associated with the diet effect for all 9730 gene pairs. We then repeated each linear mixed effects model 100 times after permuting diet, extracted the p-value associated with the diet effect, and used these values to calculate an empirical FDR distribution (*Snyder-Mackler et al., 2016*).

Using this approach, we identified 445 gene pairs that were significantly differentially co-expressed as a function of diet at a 20% empirical FDR. Next, we performed two follow-up analyses to understand their biological import. First, we tested for the existence of 'hub genes', defined as genes that displayed differential co-expression to their tested partner genes more so than expected by chance. To define the null distribution for identifying hub genes, we randomly sampled 445 gene pairs from the set of all 9730 tested gene pairs 1000 times and calculated the number of partners a focal gene had in each sample; we considered a gene to be a significant 'hub gene' if it fell outside the 95th percentile of this distribution, which was equivalent to a focal gene that displayed significant differential co-expression with 13 or more of its tested partner genes. Second, we asked whether the set of 'hub genes' we identified were enriched for transcription factors, relative to the background set of all 140 genes tested for differential co-expression. We performed this analysis because many of the proposed mechanisms to generate large scale changes in gene co-expression patterns involve changes in transcription factor function or activity (*de la Fuente, 2010*; *Gaiteri et al., 2014*). To implement the enrichment analysis, we used the TRRUST database of known mammalian transcription factors for annotation (*Han et al., 2018*) paired with hypergeometric tests.

## Weighted gene co-expression network analysis

We employed the commonly used approach of weighted gene co-expression network analysis (WGCNA) to identify and characterize modules of co-expressed genes. We used the *WGCNA* R package (*Langfelder and Horvath, 2008*) with a minimum module size of 30 genes and minimum module dissimilarity threshold of 0.25 to identify co-expression modules, which were then used for downstream analyses.

## Mediation analysis

To explore relationships between DAB score and differential gene expression, we conducted mediation analyses using a bootstrapping approach involving 10,000 bootstrap iterations of two models: (Model 1) the expression of each gene as a function of diet, and (Model 2) the expression of each gene as a function of diet and DAB score (*Preacher and Hayes, 2004*). For each bootstrap iteration, we then calculated the mediation effect (i.e. the indirect effect) of DAB score as the difference between the effect size of diet in Model 1 ($\beta_{diet}$) and Model 2 ($\beta'_{diet}$). We considered there to be a mediation effect when the 90% confidence interval for the indirect effect ($\beta_{diet}$-$\beta'_{diet}$) did not include zero.

A similar method was used to calculate the mediation of gene expression on DAB, testing the difference between the effect size of diet in two models: (Model 3) DAB as a function of diet, and (Model 4) DAB as a function of diet and the expression of each gene.

## Acknowledgements

We thank Beth Uberseder, Maryanne Post, JD Bottoms, Edison Floyd, Jason Lucas, Joshua Long, Diane Wood, and Sherri Samples for technical support. We thank Nicholas Lozier, Tiffany Pan, Marina Watowich, and Jenny Tung for their helpful feedback on previous versions of this manuscript. Funding: This work was funded by R01HL087103 (CAS), R01HL122393 (TCR), U24DK097748 (TCR) from NIH and intramural funding from the Department of Pathology, Wake Forest School of Medicine (CAS). NSM was supported by R00AG051764 and R01AG060931 from NIH, and AJL was supported by a postdoctoral fellowship from the Helen Hay Whitney Foundation. The Wake Forest Comprehensive Cancer Center Cancer Genomics Shared Resource is supported by P30CA012197 and by a NIH Shared Instrumentation Grant S10OD023409 to GAH.

## Additional information

### Funding

| Funder | Grant reference number | Author |
|---|---|---|
| National Heart, Lung, and Blood Institute | R01HL087103 | Carol A Shively |
| National Heart, Lung, and Blood Institute | R01HL122393 | Thomas C Register<br>Carol A Shively |
| National Institute of Diabetes and Digestive and Kidney Diseases | U24DK097748 | Thomas C Register |
| National Institute on Aging | R00AG051764 | Noah Snyder-Mackler |
| National Institute on Aging | R01AG060931 | Noah Snyder-Mackler |
| Wake Forest School of Medicine | | Carol A Shively |
| Helen Hay Whitney Foundation | | Amanda J Lea |
| National Cancer Institute | P30CA012197 | Timothy D Howard<br>Gregory A Hawkins<br>Thomas C Register |
| National Institutes of Health | S10OD023409 | Gregory A Hawkins |

The funders had no role in study design, data collection and interpretation, or the decision to submit the work for publication.

### Author contributions

Corbin SC Johnson, Data curation, Formal analysis, Visualization, Methodology, Writing - original draft, Writing - review and editing; Carol A Shively, Thomas C Register, Conceptualization, Resources, Data curation, Formal analysis, Supervision, Funding acquisition, Visualization, Methodology, Writing - original draft, Project administration, Writing - review and editing; Kristofer T Michalson, Ryne J DeBo, Timothy D Howard, Susan E Appt, Methodology, Writing - review and editing; Amanda J Lea, Formal analysis, Visualization, Writing - original draft, Writing - review and editing; Gregory A Hawkins, Yongmei Liu, Charles E McCall, David M Herrington, Edward H Ip, Writing - review and editing; Noah Snyder-Mackler, Data curation, Formal analysis, Supervision, Visualization, Methodology, Writing - original draft, Writing - review and editing

### Author ORCIDs

Corbin SC Johnson ![ORCID] https://orcid.org/0000-0001-9616-5755
Carol A Shively ![ORCID] https://orcid.org/0000-0002-1536-2288
Kristofer T Michalson ![ORCID] https://orcid.org/0000-0002-1927-6440
Amanda J Lea ![ORCID] https://orcid.org/0000-0002-8827-2750
Timothy D Howard ![ORCID] https://orcid.org/0000-0003-2518-4902
Susan E Appt ![ORCID] https://orcid.org/0000-0003-4953-2940
Yongmei Liu ![ORCID] https://orcid.org/0000-0002-6562-1858
Charles E McCall ![ORCID] https://orcid.org/0000-0002-9082-469X
Edward H Ip ![ORCID] https://orcid.org/0000-0002-4811-4205
Thomas C Register ![ORCID] https://orcid.org/0000-0002-4078-0166
Noah Snyder-Mackler ![ORCID] https://orcid.org/0000-0003-3026-6160

### Ethics

Animal experimentation: All animal manipulations were performed according to the guidelines of state and federal laws, the US Department of Health and Human Services, and the Animal Care and Use Committee of Wake Forest School of Medicine (ACUC #A12-195; #A15-180).

Decision letter and Author response
Decision letter https://doi.org/10.7554/eLife.68293.sa1
Author response https://doi.org/10.7554/eLife.68293.sa2

## Additional files

### Supplementary files

• Supplementary file 1. Supplementary tables 1-6. Suppl.tab 1A Effect of diet on gene expression. Suppl.tab 1B Effects of diet on gene expression (FDR < 0.05 β sort). Suppl.tab 2A Biological processes enriched in western genes compared to other measured genes Suppl.tab 2B. Biological processes enriched in mediterranean genes compared to other measured genes. Suppl. tab 3. Transcription factor binding site motif enrichment. Suppl.tab 4A. Diet correlation and gene set enrichment of weighted gene co-expression network analysis (WGCNA) co-expression modules. Suppl.tab 4B. Biological processes enriched in WGCNA co-expression modules. Suppl.tab 5A. Gene pair correlations across and within diet groups. Suppl.tab 5B. Differentially Correlated Genes. Suppl. tab 6A. Biological processes enriched in behavior-mediated differentially-expressed genes (DEGs) Suppl.tab 6B. Biological processes enriched in behavior-mediated western genes Suppl.tab 6C. Biological processes enriched in behavior-mediated mediterranean genes.

• Transparent reporting form

### Data availability

Sequencing data have been deposited in GEO under accession code GSE144314. Code can be found here: https://github.com/cscjohns/diet_behavior_immunity (copy archived at https://archive. softwareheritage.org/swh:1:rev:8f750d2bd2afc7bd12844aedf402519ea117930a).

The following dataset was generated:

| Author(s) | Year | Dataset title | Dataset URL | Database and Identifier |
|---|---|---|---|---|
| Johnson CS, Shively CA, Michalson KT, Lea AJ, DeBo RJ, Howard TD, Hawkins GA, Appt SE, Liu Y, McCall CE, Herrington D, Register TC, Snyder-Mackler N | 2021 | Contrasting Effects of Western vs. Mediterranean Diets on Monocyte Inflammatory Gene Expression and Social Behavior in a Primate Model | https://www.ncbi.nlm. nih.gov/geo/query/acc. cgi?acc=GSE144314 | NCBI Gene Expression Omnibus, GSE144314 |

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

## Appendix 1

### Regarding rank and RNA integrity (RIN)

Because of the well-established effects of social status on atherosclerosis (*Addo et al., 2012*; *Hallman et al., 2001*; *Rosengren et al., 2004*; *Shively et al., 1990*; *Steptoe and Kivimäki, 2012*, *Steptoe and Kivimäki, 2013*; *Stuller et al., 2012*; *Yusuf et al., 2004*), inflammation (*Kiecolt-Glaser, 2010*; *Maes et al., 1998*; *Steptoe et al., 2007*), and immune cell gene regulation (*Brydon et al., 2005*; *Chen et al., 2009*, *Chen et al., 2011*; *Cole, 2013*; *Miller et al., 2008*; *Snyder-Mackler et al., 2016*; *Tung et al., 2012*; *Tung and Gilad, 2013*), one of the goals of this study was to examine of social status interacted with diet to alter monocyte gene regulation. Specifically, we hypothesized that the promotion of regulatory polarization from the Mediterranean diet would attenuate the deleterious effects of social subordination. As with many RNA-sequencing data, the RNA integrity (RIN) in our data set was correlated with the first axis of variance in a PCA analysis of gene expression (Pearson's r = 0.41, p = 0.020; *Appendix 1—figure 1A*) and thus needed to be controlled for. Unfortunately, in this dataset, RIN was significantly correlated with dominance (Pearson's r = 0.49, p = 5.1 x 10-3; *Appendix 1—figure 1B*). Thus, when we controlled for RIN prior to downstream analysis, we also removed our ability to detect the effect of dominance rank on gene expression. Our subsequent analyses, and sampling at later timepoints in the study, led us to conclude this correlation was purely due to random and unidentified technical reasons (and that dominance rank does not influence RNA integrity). Samples collected at other timepoints in this study will allow us to address the potential interactions between diet and social status.

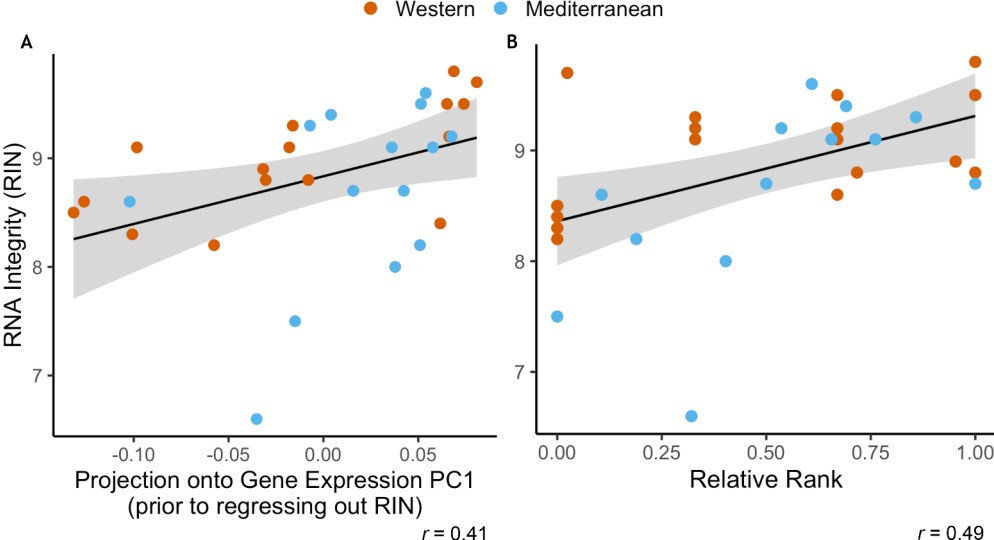

**Appendix 1—figure 1.** RNA Integrity was correlated with both uncorrected gene expression and relative rank. (**A**) RNA integrity (RIN) was correlated with PC1 of gene expression (62% of overall variance in gene expression) prior to correction for batch effects (Pearson's *r* = 0.41, *p* = 0.020). Because of this, RIN was included as a batch effect prior to downstream analysis. (**B**) RIN was also correlated with relative dominance rank (Pearson's *r* = 0.49, *p* = 5.1 x $10^{-3}$). Points are colored to indicate Western (orange) or Mediterranean (blue) diet to show that diet did not have an interactive effect in either case.

## Appendix 2

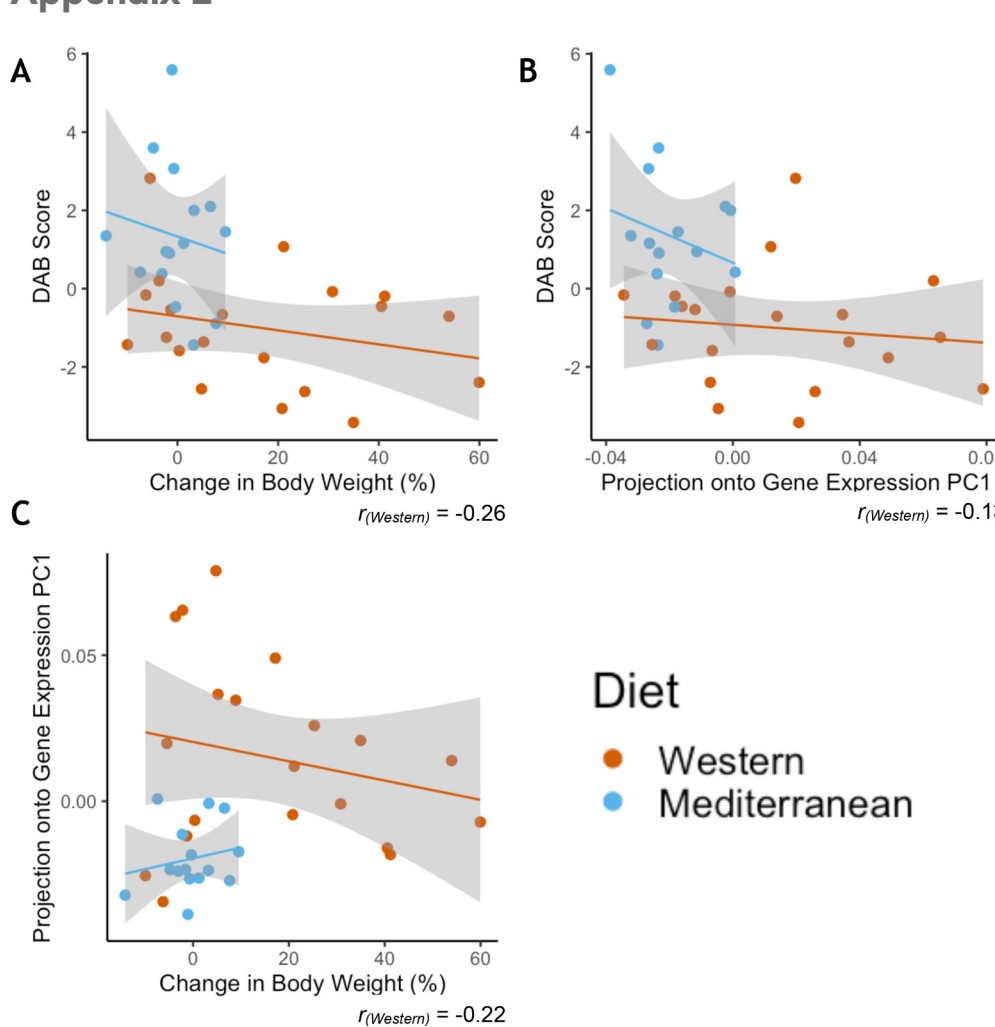

**Appendix 2—figure 1.** Greater phenotypic variability in Western diet fed monkeys does not show consistency in individual responsiveness across phenotypes. (**A**) Monkeys fed the Western diet showed more variability than monkeys fed the Mediterranean diet in both diet-altered behavior (DAB) and change in body weight. However, the two phenotypes were not correlated within monkeys fed the Western diet (Pearson's $r_{Western} = -0.26$, $p = 0.28$). (**B**) Western fed monkeys also showed more variability in the first principal component (PC1) of gene expression than Mediterranean fed monkeys. DAB and PC1 of gene expression were not significantly correlated in Western fed monkeys (Pearson's $r_{Western} = -0.13$, $p = 0.60$). (**C**) PC1 of gene expression and change in body weight were not significantly correlated in Western fed monkeys (Pearson's $r_{Western} = -0.22$, $p = 0.36$).

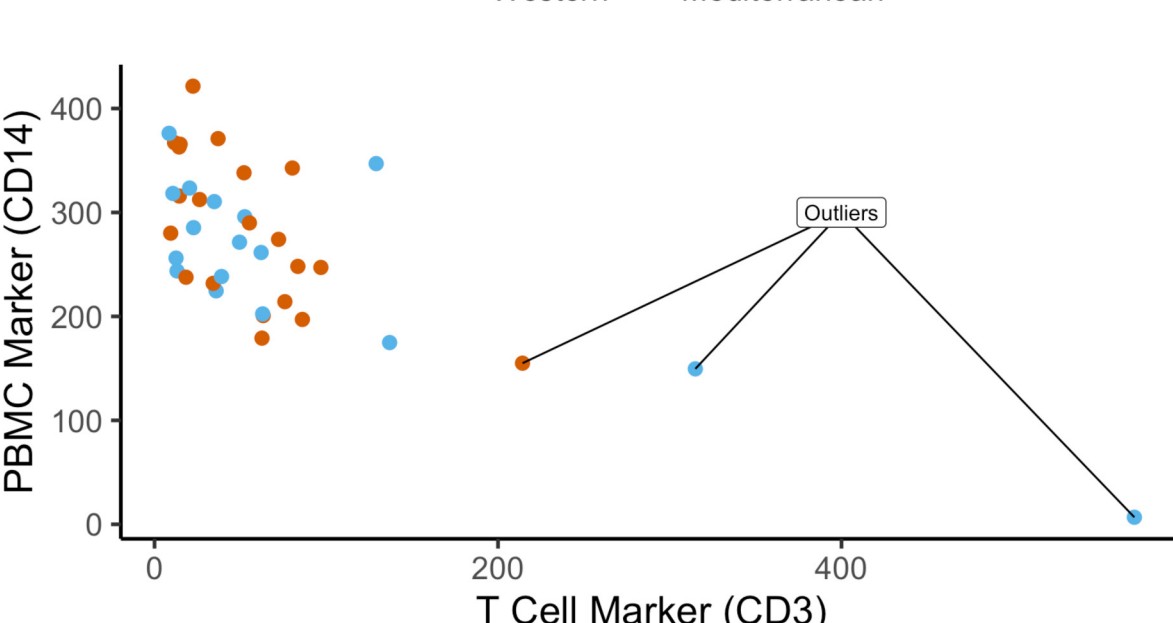

**Appendix 2—figure 2.** Quality control of cell purity by CD14 and CD3 expression levels: three samples were excluded due to lower CD14 and high CD3 – possible T cell contamination. Normalized expression (reads per kilobase million) of CD14 and CD3 are plotted as markers of monocytes and T cells, respectively. Three samples were excluded as outliers due to possible T cell contamination.

