## [Decision Letter]

**Acceptance summary:**

The authors have done a fantastic job in successfully addressing the concerns raised by the reviewers. As they report this exciting finding, more careful interpretations and detailing of the caveats have significantly strengthened the manuscript. Overall, the discussion reads much more streamlined now.

**Decision letter after peer review:**

Thank you for submitting your article "Contrasting Effects of Western v *S. mediterranean* Diets on Monocyte Inflammatory Gene Expression and Social Behavior in a Primate Model" for consideration by *eLife*. Your article has been reviewed by 3 peer reviewers, one of whom is a member of our Board of Reviewing Editors, and the evaluation has been overseen by George Perry as the Senior Editor. The following individuals involved in review of your submission have agreed to reveal their identity: Kaixiong Ye (Reviewer #1); Christina Bergey (Reviewer #2).

Essential Revisions:

While all the reviewers unanimously agree that this is an important study with sound methods, assays, and analyses, there are some serious concerns raised about the over-interpretation of results. Specifically, while the authors repeatedly cite the human relevance, we are unclear how far these results from captive primates with no shared evolutionary history with human diets, are applicable to humans. Lack of control macaques maintained in their standard diet makes it further difficult to estimate the effects of a dietary switch to novel human-like diet relative to the costs and benefits of Western vs Mediterranean diets. Are these methods and data enough to support the evolutionary mismatch hypothesis as has been claimed by the authors? Some of these issues might be addressed by redrafting certain sections (see comments below) and adding further analyses (see Reviewer 2).

*Reviewer #1 (Recommendations for the authors):*

In Abstract "uncover new pathways through which Western diets generate inflammation and disease" is an overstatement because there are no results supporting a direct link between monocyte gene expression changes and inflammation or disease.

A brief description of the study process should be provided at the beginning of the study section. Currently, all the details are in the Methods section. For instance, the sample size in each diet group is critical for the understanding of this study.

Line 561-564: "To examine global patterns of variation in gene expression, we conducted principal component analysis on the correlation matrix of normalized residual gene expression using the prcomp function in R." Principal components of gene expression, described in the Results section, are different from principal components of the correlation of gene expression, described here in the Methods section. Is this a misstatement in the Methods section? Otherwise, it is more appropriate to redo the analysis with principal components of gene expression.

The numbers in Figure 5B and 5C match the numbers in the main text, but do not match those in the figure legend. Why?

Line 356: "HICs", it is a bit confusing here. At least, the acronym is not necessary because it was only used once throughout this manuscript.

*Reviewer #2 (Recommendations for the authors):*

My recommendations for strengthening the work are minor, besides those outlined above to include caveats concerning the differences between macaques and humans that will hopefully prevent lay readers from over-interpreting the results. Specifically, species-level differences which warrant mention include gross differences in "natural" diet between the species, as well as known recent selection on diet-related genes in humans (reviewed in, e.g., Luca et al., 2010; doi:10.1146/annurev-nutr-080508-141048) and gut microbiome differences between the species (e.g., Chen et al., 2018; doi:10.1038/s41598-018-33950-6).

A simple analysis that begins to address this point analytically would be to compare what results exist for humans (e.g., Camargo et al., 2012; doi:10.1017/S0007114511005812) to those of your study. Additionally, one could check whether the DE genes you identify are known to be selected in humans. However, I know I am committing the cardinal reviewer sin of suggestion additional analyses here.

*Reviewer #3 (Recommendations for the authors):*

Authors provide compelling evidence for dietary mismatch increasing the risk of chronic diseases, by using a whole diet manipulation experiment in a non-human primate model. They performed a solid suite of behavioural assays and transcriptome analysis of specific immune cells as a proxy for physiological effects of the Mediterranean vs Western diets to mimic the human diet prevalent in a traditional hunter-gatherer society and the modern western world respectively. Their interpretation of dietary effects on gene expression in monocyte populations and immune cell polarization (pro-inflammatory vs regulatory Monocyte cells), correlated gene expression, identification of hub genes was convincing and quite thoughtful. Finally, the use of mediation analysis to propose how both differential immune gene expression and behavioural changes might influence their respective outcomes of dietary changes was appropriate and opens up avenues for future research. Overall, the manuscript is well-written and delivers the message clearly.

However, my major concern is the suitability of these results to explain human relevance and how far they can address the actual evolutionary significance. I think they should tone down a little. For example, is there really any strong reason to assume that macaques will mimic dietary responses in humans? I appreciate the fundamental importance of macaque-specific responses, but I am unclear how captive primates can model human effects─ how do authors factor their (obvious?) fundamental differences between different immune response profiles activated against similar cues and standing microbiome, warranting divergent interactions with the said dietary manipulations. I think these are caveats that need to be carefully discussed as early as possible (e.g. briefly in abstract and results, and certainly in the first paragraph of the discussion) to avoid building over expectations among readers.

This is slightly unfortunate because there is no full control treatment where macaques are maintained in their regular diet (i.e., standard monkey chow) and then compared with groups switched to the Mediterranean vs western diet to estimate the relative deviations from their expected physiological processes and behavioural traits. I think this limitation must be highlighted as much as possible.

Could there be more discussion on the relevance of differentially expressed macaque genes in humans?

What are the possible fates of other immune pathways after dietary manipulations? It will be helpful to add some brief speculations.

---

## [Author Response]

1. Limitations of extrapolation to human health and disease.– From Essential Revisions: Specifically, while the authors repeatedly cite the human relevance, we are unclear how far these results from captive primates with no shared evolutionary history with human diets, are applicable to humans.– From Reviewer 3: However, my major concern is the suitability of these results to explain human relevance and how far they can address the actual evolutionary significance. I think they should tone down a little. For example, is there really any strong reason to assume that macaques will mimic dietary responses in humans? I appreciate the fundamental importance of macaque-specific responses, but I am unclear how captive primates can model human effects─ how do authors factor their (obvious?) fundamental differences between different immune response profiles activated against similar cues and standing microbiome, warranting divergent interactions with the said dietary manipulations. I think these are caveats that need to be carefully discussed to avoid building over expectations among readers.– From Reviewer 3: Could there be more discussion on the relevance of differentially expressed macaque genes in humans?

We appreciate the concern regarding possible overinterpretation of results. There is an extensive body of literature demonstrating the utility of the cynomolgus macaque model to explore influences of diet on numerous phenotypes including atherosclerosis and cardiovascular disease, bone metabolism, breast and uterine biology, and other phenotypes (Adams et al., 1997; Clarkson et al., 2004, 2013; Cline et al., 2001; Cline and Wood, 2006; Haberthur et al., 2010; Lees et al., 1998; Mikkola et al., 2004; Mikkola and Clarkson, 2006; Naftolin et al., 2004; Nagpal, Shively, et al., 2018; Nagpal, Wang, et al., 2018; Register, 2009; Register et al., 2003; Shively and Clarkson, 2009; Sophonsritsuk et al., 2013; Walker et al., 2008; Wood et al., 2007). The cynomolgus model was remarkably accurate in predicting effects of hormone therapies on both cardiovascular disease and breast cancer later demonstrated in the very large Women’s Health Initiative (Adams et al., 1997; Clarkson et al., 2013; Naftolin et al., 2004; Shively and Clarkson, 2009; Wood et al., 2007). Cynomolgus macaque responses to other therapies (tamoxifen, selective estrogen receptor modulators, blood pressure medications, etc.) also have shown great similarities to those in humans (Cline et al., 2001). We have added additional text to the Abstract (lines 51-52), Introduction (lines 136-141), and Discussion (lines 531-542) to situate the current work in the extensive literature that uses cynomolgus macaques as a model to understand human health. We have also included discussion regarding the limitations of extrapolating these results to humans in lines 543-545 of the Discussion

We also tested the overlap of differential gene expression induced by the Western diet with genes implicated in human complex traits (Zhang et al., 2020). Genes implicated in numerous traits associated with cardiometabolic health were enriched in Western genes, while no traits were enriched in Mediterranean genes. We describe these findings in lines 206-215 of the Results section and in Figure 1—figure supplement 1, which depicts traits relevant to human health and disease identified by previous groups where gene expression profiles overlapped with the “Western genes” in the current study. Lines 668-672 of the Materials and methods detail the statistical approach used.

2. Limitations of this experimental design to test the evolutionary mismatch hypothesis.– From Reviewer 2: My recommendations for strengthening the work are minor, besides those outlined above to include caveats concerning the differences between macaques and humans that will hopefully prevent lay readers from over-interpreting the results. Specifically, species-level differences which warrant mention include gross differences in "natural" diet between the species, as well as known recent selection on diet-related genes in humans (reviewed in, e.g., Luca et al., 2010; doi:10.1146/annurev-nutr-080508-141048) and gut microbiome differences between the species (e.g., Chen et al., 2018; doi:10.1038/s41598-018-33950-6).– From Reviewer 2: A simple analysis that begins to address this point analytically would be to compare what results exist for humans (e.g., Camargo et al., 2012; doi:10.1017/S0007114511005812) to those of your study.– From Reviewer 2: Additionally, one could check whether the DE genes you identify are known to be selected in humans.

We appreciate the suggestion to strengthen our discussion of the macaque model of human health. As with early hunter-gatherer humans, macaques are omnivorous in the wild, eating a variety of plants and animals. In addition, the cynomolgus macaque often co-exists with human populations, and in that respect may have co-evolved in many ways. Furthermore, cynomolgus macaques have been used in studies of dietary influences on chronic prevalent human disease for 50 years (Malinow et al., 1972), and nearly 700 papers in a Pubmed literature search support the idea that cynomolgus responses to diet are remarkably similar to those of humans in all systems studied. Some of these studies are identified above. With respect to the microbiome, previous work by others has demonstrated that the gut microbiome of omnivorous nonhuman primates is similar to that of humans living a modern lifestyle (Ley et al., 2008), and we previously reported similarities in patterns of microbiome responses to Mediterranean vs. Western diets between humans and NHPs in the present study (Nagpal, Shively, et al., 2018). We have added discussion of the above and note limitations of extrapolation to humans due to species-level differences in natural diets and the role that selection may plan in responses of humans to Western or Mediterranean dietary patterns (lines 543-545). Similarities between humans in DE genes are noted in responses above. In addition, we already had noted that our studies complement and extend the findings of Camargo (line 399), and we added more detail that we found similar effects of diet on expression of IL6 and NF-κB pathway members (line 397).

3. Lack of control group maintained on a standard chow diet.– From Essential Revisions: Lack of control macaques maintained in their standard diet makes it further difficult to estimate the effects of a dietary switch to novel human-like diet relative to the costs and benefits of Western vs Mediterranean diets.– From Reviewer 3: This is slightly unfortunate because there is no full control treatment where macaques are maintained in their regular diet (i.e., standard monkey chow) and then compared with groups switched to the Mediterranean vs western diet to estimate the relative deviations from their expected physiological processes and behavioural traits.

We appreciate the concern regarding the lack of a standard monkey chow diet control group. All monkeys ate chow during the baseline phase and were thoroughly phenotyped, exhibiting minimal differences in monocyte gene expression profiles between groups subsequently assigned to the two diets, which involved stratified randomization based on key baseline characteristics while consuming the same diet. Importantly, monkey chow is unlike any historic or current human or nonhuman primate diet as is apparent in Table 1. It is quite low in fat, and rich in soy protein and isoflavones, which are known to alter physiology and immune system function. Therefore, parallel assessments of health measures in monkeys consuming chow long term do not provide data relevant to diet effects on human health. We have added discussion of the strengths of the study (lines 136-141, 531-542), which was designed in order to be able to draw causal inference about the diet manipulation, and we acknowledge limitations to assess directionality of changes (i.e. which experimental diet is driving a particular observed difference) in lines 545-553.